# Erythrocytosis-inducing PHD2 mutations implicate biological role for N-terminal prolyl-hydroxylation in HIF1α oxygen-dependent degradation domain

Cassandra C Taber[1], Wenguang He[1,2], Geneviève MC Gasmi-Seabrook[3], Mia Hubert[1], Fraser G Ferens[1], Mitsuhiko Ikura[3,4], Jeffrey E Lee[1], Michael Ohh[1,2]*

[1]Department of Laboratory Medicine and Pathobiology, Temerty Faculty of Medicine, University of Toronto, Toronto, Canada; [2]Department of Biochemistry, Temerty Faculty of Medicine, University of Toronto, Toronto, Canada; [3]Princess Margaret Cancer Centre, University Health Network, Toronto, Canada; [4]Department of Medical Biophysics, Temerty Faculty of Medicine, University of Toronto, Toronto, Canada

## eLife Assessment

In this **valuable** study, Taber et al. used a battery of biophysical and structural approaches to characterize the impact of erythrocytosis-related mutations in prolyl hydroxylase domain protein 2 (PHD2). The authors show that PHD2 mutant proteins are destabilized, thus supporting the tenet that dysregulation of PHD2/hypoxia induced factor (HIF) axis underpins erythrocytosis, while providing **solid** evidence that N-terminal ODD prolyl hydroxylation of HIF is indispensable for these phenotypes. These findings were found to be of interest for researchers focusing on oxygen sensing in homeostasis and pathological states.

*For correspondence:
michael.ohh@utoronto.ca

Competing interest: The authors declare that no competing interests exist.

**Abstract** Mutations in *EGLN1*, the gene encoding for hypoxia-inducible factor (HIF) prolyl-4-hydroxylase 2 (PHD2), cause erythrocytosis and in rare cases the development of neuroendocrine tumors. In the presence of oxygen, PHD2 hydroxylates one or both conserved prolines in the oxygen-dependent degradation domain (ODD) of HIFα subunits, sufficiently marking HIFα for binding and ubiquitylation via the von Hippel-Lindau (VHL) tumor suppressor protein-containing E3 ubiquitin ligase and subsequent degradation by the 26S proteasome. However, prolyl-hydroxylation in the C-terminal ODD appears to be the predominant and sufficient event in triggering the oxygen-dependent destruction of HIFα, rendering the biological significance of N-terminal ODD proline unclear. Here, we examined seven disease-associated *EGLN1* mutations scattered across the catalytic core and showed definitively that all PHD2 mutants have a structural and/or catalytic activity defect as measured by time-resolved nuclear magnetic resonance. Notably, we identified one of the PHD2 mutants, P317R, to retain comparably wild-type capacity to hydroxylate the predominant proline in the C-terminal ODD but had uniquely compromised ability to hydroxylate the N-terminal ODD proline. These findings support the notion that deregulation of HIF ultimately underlies PHD2-driven erythrocytosis and challenge the currently held uncertainty that the N-terminal ODD prolyl-hydroxylation event is dispensable in normal hypoxic signaling pathway.

## Introduction

About half of blood volume is made up of erythrocytes (red blood cells) whose role is to deliver oxygen from the lungs to the tissues in the body (*Lee and Percy, 2011*). Low concentrations of red blood cells result in insufficient oxygen delivery to tissues, while high concentrations can induce a propensity toward thrombotic events, rendering red blood cell mass regulation crucial. Erythropoietin (EPO) is a hormone produced primarily in the kidneys that stimulates erythropoiesis (*Weidemann et al., 2009*; *Richmond et al., 2005*). Through binding to EPO receptors on erythroid progenitors, EPO can activate a signaling cascade that induces differentiation into erythrocytes (*Bhoopalan et al., 2020*). EPO production is regulated in response to physiological oxygen concentrations through hypoxia response elements (HREs). HREs are enhancers that transcription factors called hypoxia-inducible factors (HIFs) bind to and subsequently upregulate the expression of numerous hypoxic response genes, including *EPO* (*Semenza and Wang, 1992*). Disrupted EPO regulation can be pathogenic and lead to an autonomous overproduction of erythrocytes causing erythrocytosis (*Mithoowani et al., 2020*). Due to EPO's inherent link to the oxygen response, mutations in the metazoan oxygen-sensing pathway have frequently been found to underlie erythrocytosis.

The oxygen-sensing pathway has been conserved in all animals and is essential for the maintenance of cellular activities in low oxygen settings (*Lippl et al., 2018*). Under hypoxia, oxygen-labile HIF1-3α subunits complex with its constitutively expressed HIFβ subunits to translocate to the nucleus as active heterodimeric transcription factors (*Dengler et al., 2014*). Following translocation, HIF binds to HREs across the genome to trigger the transactivation of numerous genes, such as *GLUT1*, *EPO*, and *VEGF*, to promote anaerobic metabolism, erythropoiesis, and angiogenesis, respectively, to induce adaptation to low oxygen availability (*Schödel et al., 2011*). In normal oxygen conditions, HIF prolyl hydroxylases (PHDs) hydroxylate HIFα subunit at one or both of the two conserved proline sites located in the oxygen-dependent degradation domain (ODD), which initiates its ubiquitylation via the von Hippel-Lindau tumor suppressor protein (pVHL)-containing E3 ubiquitin ligase (Elongin BC/Cullin 2-Rbx1/pVHL) and subsequent destruction via the 26S proteasome. Notably, only one proline is required to be hydroxylated to trigger its rapid destruction, and whether HIFα is singly or doubly hydroxylated does not appear to change the rate of its ubiquitylation or proteasome-mediated destruction (*He et al., 2021*). Moreover, the C-terminally located proline within ODD or CODD (e.g. P564 on HIF1α) has been shown to be the predominant residue that becomes hydroxylated prior to the modification of proline in the N-terminal ODD or NODD (e.g. P402 on HIF1α) (*He et al., 2021*; *Chan et al., 2005*; *He et al., 2024*). The reason for this preferential prolyl-hydroxylation is incompletely understood, and the biological significance or the mechanistic purpose of the N-terminal proline remains unclear.

Clinically observed mutations have been identified in the major components of the oxygen-sensing pathway, including *VHL* (pVHL) (*Clifford and Maher, 2001*; *Neumann and Wiestler, 1991*), *EPAS1* (HIF2α) (*Tarade et al., 2018*; *Pacak et al., 2013*; *Pacak et al., 2014*), *EGLN1* (PHD2) (*Delamare et al., 2023*; *Gardie et al., 2014*), and *EGLN2* (PHD1) (*Yang et al., 2015*). These have been associated with the development of diseases with overlapping phenotypes, with erythrocytosis being the most common. Collectively, these diseases are referred to as pseudohypoxic diseases as they involve an inappropriate hypoxic response. Erythrocytosis caused by mutations in *EGLN1* is the most recently discovered pseudohypoxic disease. Since its first reported case in 2006 (*Percy et al., 2006*), 152 cases with 96 distinct mutations have been reported (*Supplementary file 1*). *EGLN1* mutation-driven erythrocytosis is referred to as erythrocytosis, familial, 3 (ECYT3) by the Online Mendelian Inheritance in Man (OMIM). However, three individuals with distinct *EGLN1* mutations (W51X, A228S, and H374R) developed recurrent pheochromocytoma and paraganglioma (PPGL) (*Yang et al., 2015*; *Provenzano et al., 2022*; *Ladroue et al., 2008*), which are neuroendocrine tumors that develop from chromaffin cells either within (pheochromocytoma) or outside (paraganglioma) the adrenal gland (*Lenders et al., 2005*).

Of the three isozymes of PHD, PHD2 is thought to be the most critical with high expression seen in a variety of cell types (*Berra et al., 2003*) and the only PHD able to effectively target both prolines in HIF1α and 2α ODDs (*Chan et al., 2005*; *Appelhoff et al., 2004*; *Flashman et al., 2008*). PHD2 belongs to the 2-oxoglutarate-dependent dioxygenase superfamily (*Schofield and Ratcliffe, 2004*) in which enzymes contain a double-stranded β-helix fold with a highly conserved HXE/D…H motif that coordinates $Fe^{2+}$ at its catalytic site (*Clifton et al., 2006*). In PHD2, the iron-coordinating residues are His313, Asp315, and His374 (*Figg et al., 2023*). Erythrocytosis-causing mutations at (H374R)

and near these conserved catalytic residues have been shown to be defective at regulating HIFα in a cellular context (*Delamare et al., 2023*; *Percy et al., 2006*; *Ladroue et al., 2008*), which appears to be a common mechanism observed in the other pseudohypoxic diseases. For example, numerous studies have revealed that pVHL mutants have varied abilities to recognize and regulate HIFα, leading to the development of PPGL, erythrocytosis, and other cancers, such as clear-cell renal cell carcinoma and hemangioblastoma (*Ohh et al., 2022*; *Rechsteiner et al., 2011*; *Maher and Kaelin, 1997*). Recently, we showed that phenotypic severity of Pacak-Zhuang syndrome, caused by *EPAS1* mutations, is correlated to the ability of HIF2α mutants to escape degradation mediated by pVHL and PHD2 (*Tarade et al., 2018*; *Ferens et al., 2024*).

Together, these observations suggest that disproportionately high levels of HIFα are responsible for pseudohypoxic disease phenotypes, such as erythrocytosis and PPGL. In keeping with this notion, disease-causing mutants of PHD2 are likely defective in catalyzing HIFα hydroxylation to some extent, resulting in less HIFα degradation and correspondingly increased hypoxia response signaling. However, the precise mechanism(s) underlying such defect in PHD2 function is unclear. Recently, we devised a time-resolved nuclear magnetic resonance (NMR) assay to directly measure the rate of hydroxylation on the two conserved proline residues simultaneously in HIF1α ODD (*He et al., 2024*). Here, we examined seven disease-associated PHD2 mutants via biophysical analyses and show that all PHD2 mutants have structural and/or catalytic activity defects. While all mutants had the anticipated defect in binding and hydroxylating the predominant CODD proline, we identified the PHD2 P317R mutant that retained comparably wild-type (WT) level of hydroxylating CODD proline (i.e. HIF1α P564) but failed to hydroxylate NODD proline (i.e. HIF1α P402) as measured by NMR. Unexpectedly, PHD2 P317R was capable of binding to both NODD and CODD peptides with similar affinity. These results provide a biophysical mechanism that, at least in part, underlies PHD2-driven erythrocytosis and serendipitously reveal direct biochemical evidence to suggest that NODD prolyl-hydroxylation is necessary for normal physiological response to hypoxia.

## Results

### Mutations across the PHD2 enzyme induce erythrocytosis

We compiled and analyzed all reported cases of PHD2-driven erythrocytosis as of January 2025. 96 distinct variants have been reported, including two missense variants commonly found in Tibetan populations that are hypothesized to be adaptations to high altitude conditions (D4E and C127S) (*Song et al., 2014*; *Xiang et al., 2013*). Consistent with erythrocytosis in the general population, most patients with PHD2-driven erythrocytosis were male (~70%). About half of the case reports provided information on family history of PHD2-driven erythrocytosis, of which ~75% were reported to be familial while ~25% were de novo mutations; a similar rate of de novo mutations is observed with VHL disease. PHD2-driven erythrocytosis is typically classified as secondary erythrocytosis, which is defined as being caused by an EPO-inducing mechanism downstream of the primary defect, ultimately resulting in normal to high EPO levels that drive erythropoiesis. This classification is aligned with the suspected mechanism of PHD2-driven disease in which mutated PHD2 dysregulates HIF leading to increased EPO production. Curiously, however, EPO was reported to be largely normal in patients with PHD2-driven erythrocytosis, with few exceptions (*Supplementary file 1*), which is consistent with the loss of PHD2 in mice leading to EPO hypersensitivity in erythroid progenitors, enabling a 'normal' amount of EPO to generate a pathogenic response (*Arsenault et al., 2013*).

All cases of PHD2-driven erythrocytosis were reported as germline heterozygous mutations of PHD2, suggesting haploinsufficiency of PHD2 is enough to induce erythrocytosis. This is consistent with previous reports that HIFα regulation is dose dependent on PHD2 knockdown via siRNA (*Berra et al., 2003*). As stated previously, in three exceptional cases, individuals with PHD2 mutations developed PPGL (W51X, A228S, and H374R). Biopsies of each tumor revealed a loss of heterozygosity for PHD2, implying a complete loss of WT PHD2 in chromaffin cells is necessary to induce tumorigenesis.

The majority of the reported mutations were missense (~74%) (*Figure 1A*). Interestingly, mutations that should theoretically severely disrupt PHD2 structure (nonsense, frameshift, and deletions) caused the same phenotype (erythrocytosis) as less disruptive missense mutations, suggesting an unclear or lack of correlation between mutation type and phenotypic severity. Thus, a potential phenotypic mechanism of PHD2-driven disease can be hypothesized where erythrocytosis will develop upon

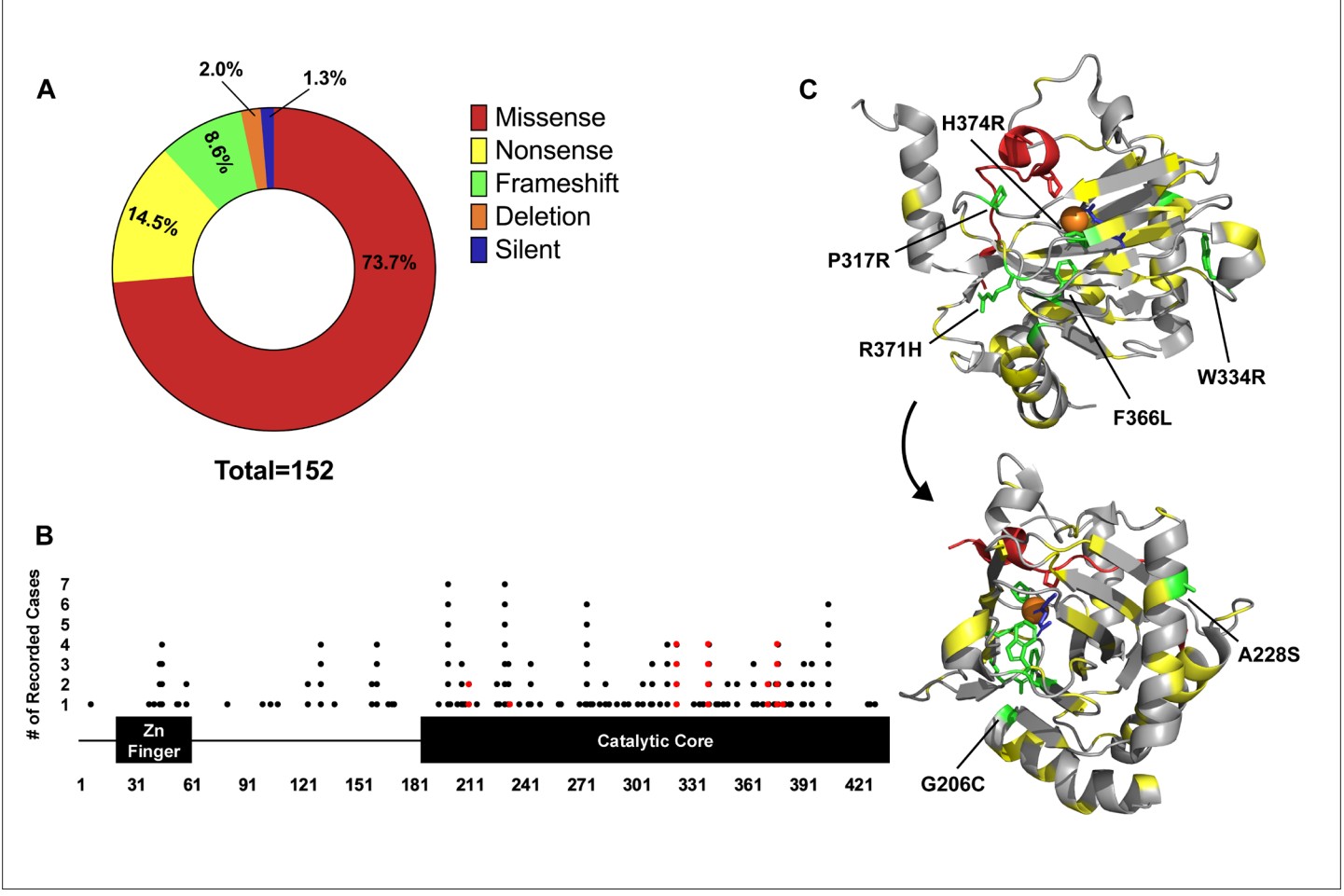

**Figure 1.** Analysis of disease-causing PHD2 mutants. (**A**) Distribution of mutation type calculated from clinical case reports. Case reports have been compiled and summarized in *Supplementary file 1B*. Linear map of mutant location and frequency on PHD2. The zinc finger is comprised of residues 21–58 while the catalytic core ranges from 181 to 426. Each dot represents a clinical report of a disease-causing mutation at the given residue. Mutations selected for analysis have been highlighted in red. (**C**) Structure of PHD2 HIF1αCODD complex (PDB:5L9B) with PHD2 mutant locations highlighted (yellow). PHD2 (gray) and HIF1αCODD (red) are depicted as ribbons. PHD2 mutants selected for analysis are highlighted in green and labeled. The structure was turned 260° on the y-axis to highlight all mutant locations.

deleterious mutation in one PHD2 allele, while PPGL will be induced upon a loss of heterozygosity in chromaffin cells.

Mutations are distributed across all exons of PHD2 with some clustering across the catalytic core (*Figure 1B*). When mapped onto the three-dimensional crystal structure of the PHD2 catalytic core, which we and others have previously solved (*Ferens et al., 2024*; *Chowdhury et al., 2016*; *McDonough et al., 2006*), these mutations do not seem to display a specific pattern (*Figure 1C*). Accordingly, representative PHD2 mutants were selected to analyze the mechanism of disease caused by PHD2 mutations. Some mutants were chosen based on their frequency in the clinical data and their presence in potential mutational hot spots. Various mutations were noted at W334 and R371 sites, while F366L was identified in multiple individuals. 9 cases of PHD2-driven disease were reported to be caused from mutations located between residues 200–210, while 13 cases were reported between residues 369 and 379. Thus, G206C and R371H were chosen to represent these potential hot spots. To examine a potential genotype-phenotype relationship, two of the mutants, A228S and H374R, responsible for neuroendocrine tumor development were also selected. Finally, mutations located close to or on catalytic core residues (P317R, R371H, and H374R) were chosen to test for suspected defects. Ultimately, we selected seven PHD2 mutants with distinct residue changes in the catalytic core of PHD2 for study.

## Some PHD2 mutants are less stable and defective in regulating HRE-driven transcription

We asked whether the enzymatic activity of PHD2 mutants can be inferred from downstream HIF activity using a dual luciferase reporter assay under control of HRE. Notably, we first generated CRISPR/Cas9-mediated PHD2 knockout (-/-) HEK293A cells to minimize the impact of endogenous PHD2 on the prolyl-hydroxylation of HIFα. As expected, WT ectopic PHD2 markedly reduced the HRE-luciferase activity driven by ectopic HIF1α compared to cells without ectopic PHD2 expression (*Figure 2A*). Two disease-associated PHD2 mutants, P317R and H374R, displayed significantly reduced ability to knock down luciferase activity (~2-fold and ~2.5-fold, respectively) compared to PHD2 WT. Notably, the other mutants showed comparable activity to PHD2 WT in downregulating HRE-driven luciferase signal under these in cellulo experimental conditions (*Figure 2A*).

We next performed a cycloheximide (CHX) chase assay to determine the protein stability of PHD2 mutants. We observed that mutants G206C, W334R, F366L, R371H, and H374R showed noticeable instability by 24 hr (*Figure 2B and C*). These results suggest that while most PHD mutants showed diminished protein stability in comparison to PHD2 WT, the stability of PHD2 mutants cannot alone explain the impact of these disease-associated mutations on HIF1α-driven HRE-luciferase activity in cells.

## PHD2 mutants are thermally unstable and some are prone to aggregation

We asked if the disease-causing mutations compromised other biophysical characteristics of PHD2. We purified catalytic cores of PHD2 harboring the selected mutations for the subsequent biophysical studies. Size exclusion chromatography (SEC) was performed as a polishing step, and chromatograms were acquired with monomeric PHD2 eluting from the column around 16.5 ml (*Figure 3A*). Mutants were eluted from the column at different volumes ranging from 16.2 ml (G206C) to 16.7 ml (P317R), while WT PHD2 was eluted consistently at 16.5 ml. These results suggest that mutations may affect the hydrodynamic size of PHD2, hinting at mutation-induced structural alteration. Moreover, PHD2 G206C, W334R, and H374R aggregated severely in SEC and throughout the purification process and could not be purified.

We next subjected the remaining four mutants (A228S, P317R, F366L, and R371H) in comparison to WT PHD2 to circular dichroism (CD) to determine structural abnormalities. The CD spectra of the mutant PHD2s did not display appreciable differences from WT PHD2 (*Figure 3B*), indicating that these PHD2 mutants' overall structure is mostly unaffected.

To further probe the stability of PHD2 mutants, molar ellipticity was measured at 220 nm from 25°C to 95°C to determine melting temperature ($T_m$). Alpha helices produce a strong minimum at 220 nm so this wavelength was selected to measure thermal protein unfolding (*Greenfield, 2006*). Melting curves were created by setting the peak value of each spectrum at 25°C to 1.0. The data were then fitted with a sigmoidal curve to produce melting curves and calculate $T_m$'s (*Figure 3C* and *Table 1*). WT PHD2's $T_m$, estimated as the temperature at which half of the proteins are unfolded, was determined to be 52.3°C. All four mutants had lower $T_m$'s than WT, suggesting a degree of thermal instability. Although it is reasonable to assume that instability would result in less PHD2, it remains unclear if the unstable enzyme is active prior to degradation and whether the remaining portion of mutant PHD2s would be sufficient to bind and hydroxylate HIFα.

## PHD2 mutants have weaker binding to HIFα than WT PHD2

Assuming intact catalytic function, the strength of PHD2 binding affinity to HIFα should translate to the extent of prolyl-hydroxylation of HIFα via PHD2. We performed microscale thermophoresis (MST), which utilizes the motion of molecules in a temperature gradient to determine dissociation constants ($K_d$), to measure the binding deficiencies, if any, between PHD2 mutants and HIFα. In addition to being highly sensitive, MST is also advantageous for measuring protein-ligand binding in an untethered setting. We showed previously that this technique was effective at distinguishing between disease classes of HIF2α mutants in Pacak-Zhuang syndrome by measuring their binding affinities to WT PHD2 (*Ferens et al., 2024*). We performed a similar protocol with PHD2 mutants to identify potential binding defects with HIF1α and 2α peptides of the C-terminal oxygen-dependent degradation domain (CODD) that contains single target proline, which is thought to be predominant over

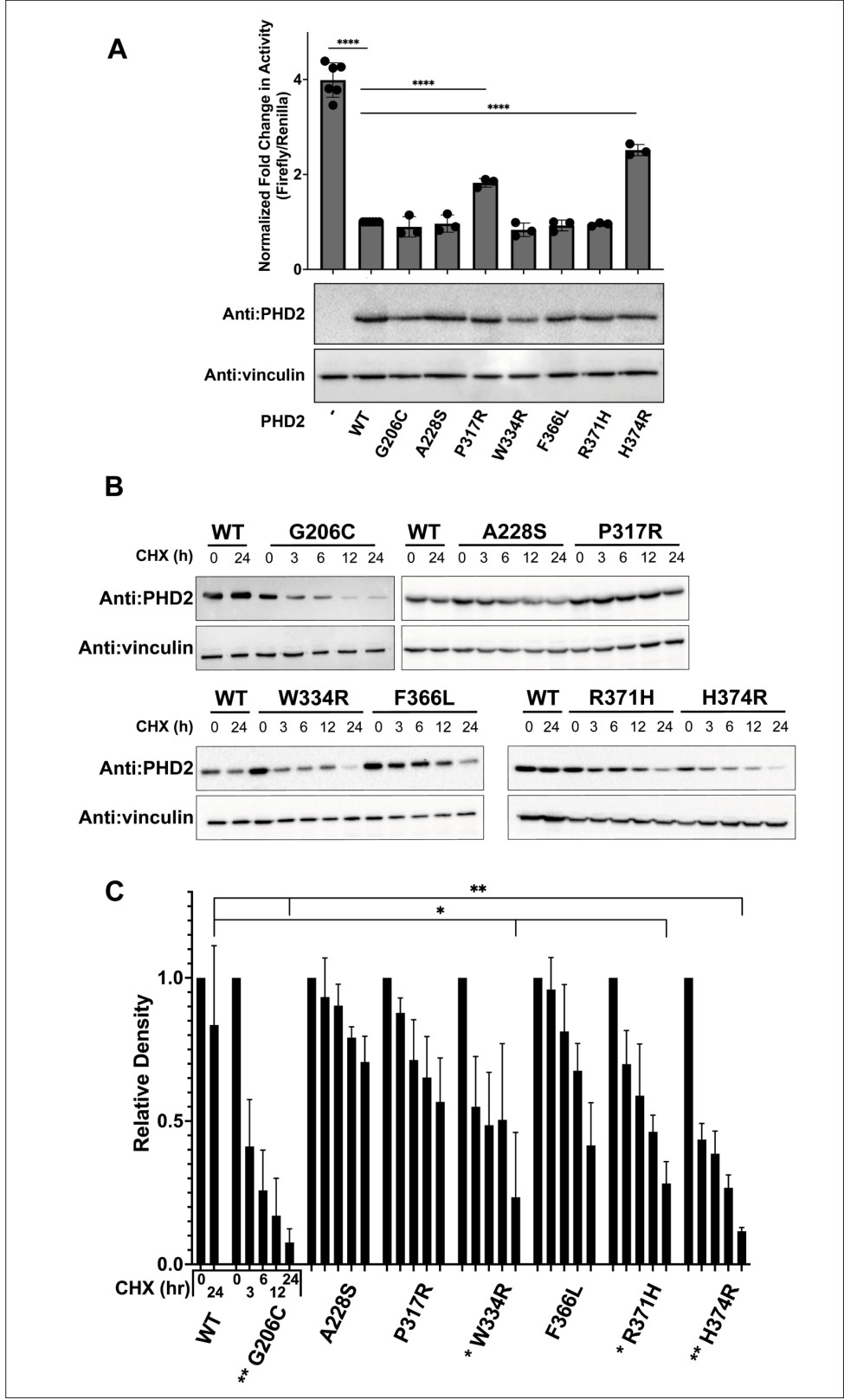

**Figure 2.** Methods in HEK293A cells detect defects in some PHD2 mutants but not all. (**A**) Dual luciferase reporter assays were performed to measure HIF1α transcriptional activity in the presence of PHD2 mutants in PHD2 -/- HEK293A cells. Individual data points are plotted. Loading accuracy was evaluated via immunoblotting for FLAG-tagged PHD2 and vinculin. (**B**) PHD2 mutant stability was measured via cycloheximide chase assay. HEK293A cells

*Figure 2 continued on next page*

*Figure 2 continued*

were transfected with wild-type or mutant PHD2 constructs, with the amount of transfected adjusted to ensure equal expression at 0 hr. After 24 hr, the transfected cells were treated with cycloheximide (CHX) to halt protein production and monitor the stability of PHD2. Cells were harvested at various time points up to 24 hr and lysates were immunoblotted to measure PHD2 levels. (**C**) FLAG immunoblot density was quantified at each time point and normalized with vinculin density to yield a relative density. 24 hr time points were compared to determine significance. For A and C, bars represent mean values, and standard error is represented by error bars (n=3, * indicates $p<0.0332$, ** indicates $p<0.0021$, *** indicates $p<0.0002$, and **** indicates $p<0.0001$ [two-tailed t-test]).

The online version of this article includes the following source data and figure supplement(s) for figure 2:

**Source data 1.** Original membranes corresponding to *Figure 2A*.

**Source data 2.** Original membrane image files corresponding to *Figure 2A*.

**Source data 3.** Original membranes corresponding to *Figure 2C*.

**Source data 4.** Original membrane image files corresponding to *Figure 2C*.

**Figure supplement 1.** Validation of PHD2 CRISPR knockout (KO) in HEK293A cells.

**Figure supplement 1—source data 1.** Original, replicate membranes run to monitor PHD2 KO.

**Figure supplement 1—source data 2.** Original membrane image files corresponding to *Figure 2—figure supplement 1*.

the N-terminal oxygen-dependent degradation domain (NODD) (*He et al., 2021*). PHD2 WT bound tightly to HIF1αCODD and HIF2αCODD peptides (7.5 and 15 µM, respectively). Notably, all but one tested PHD2 mutants (A228S, F366L, and R371H) displayed mild binding defects, up to approximately twofold, to HIF1αCODD in comparison to WT PHD2, while P317R showed a severe binding defect with high $K_d$ value (320 µM) (*Figure 4A* and *Table 2*). A similar pattern of binding deficiencies was noted for PHD2 mutants to HIF2αCODD (*Figure 4B* and *Table 2*), suggesting no significant preferential binding defects to either HIFα paralog.

## NMR reveals major defect in PHD2 P317R-mediated hydroxylation of NODD proline

Previously, we developed an assay utilizing NMR to directly measure PHD2 hydroxylation of the critical two prolines located within the intrinsically disordered ODD of HIF1α (*He et al., 2024*). Under the catalysis of PHD2, movement of resonances corresponding to the target prolines, as well as adjacent residues, is observed, reflecting local conformational changes due to the post-translational modification of HIF1αODD. The time-dependent shift in peak intensity from unhydroxylated to hydroxylated species can be interpreted as a function of the reaction progress. Notably, the enzymatic kinetics of the hydroxylation of both target prolines (P402 in NODD, P564 in CODD) can be measured simultaneously over time by using the resonance movement of neighboring residues as reporters (A403 for NODD, I566 for CODD; *Figure 5—figure supplement 1*). Since binding experiments with PHD2 mutants did not implicate marked preferential defects for either HIF1α or HIF2α, these assays were performed with HIF1αODD, of which more than 95% of amino acids have been assigned to their corresponding 3D NMR backbone cross-peak (*He et al., 2024*).

We selected PHD2 A228S for NMR study to determine if this mutant, which showed negligible to minute defect in the above studies from HRE-driven luciferase assay to MST-based binding experiments, exhibited any appreciable enzymatic defect in hydroxylating P564 or P402. We included as controls PHD2 WT and P317R that showed major defects in binding HIF1αCODD peptide and transcriptional activation of HRE-luciferase. As shown previously (*He et al., 2024*), PHD2 WT showed robust and rapid hydroxylation of P564 in CODD and a markedly slower hydroxylation of P402 in NODD (*Figure 5A*). PHD2 A228S mutant effectively hydroxylated P564 faster than P402 with negligible differences in the enzymatic rate to PHD2 WT (*Figure 5A*). Unexpectedly, P317R displayed a near WT hydroxylation of P564 but failed to hydroxylate P402 (*Figure 5A*). Superimposition of $^{1}$H-$^{15}$N HSQC spectra showed that the cross-peak of P564 shifts with PHD2 WT, A228S, and P317R mutant, but P402 only shifts when incubated with PHD2 WT and A228S, not PHD2 P317R (*Figure 5B*). These results are inconsistent with the observations made via MST that showed P317R having a severe binding defect for HIF1αCODD peptide. Furthermore, it is of particular interest to investigate whether the lack of activity of PHD2 P317R on NODD is due to a loss of binding interaction.

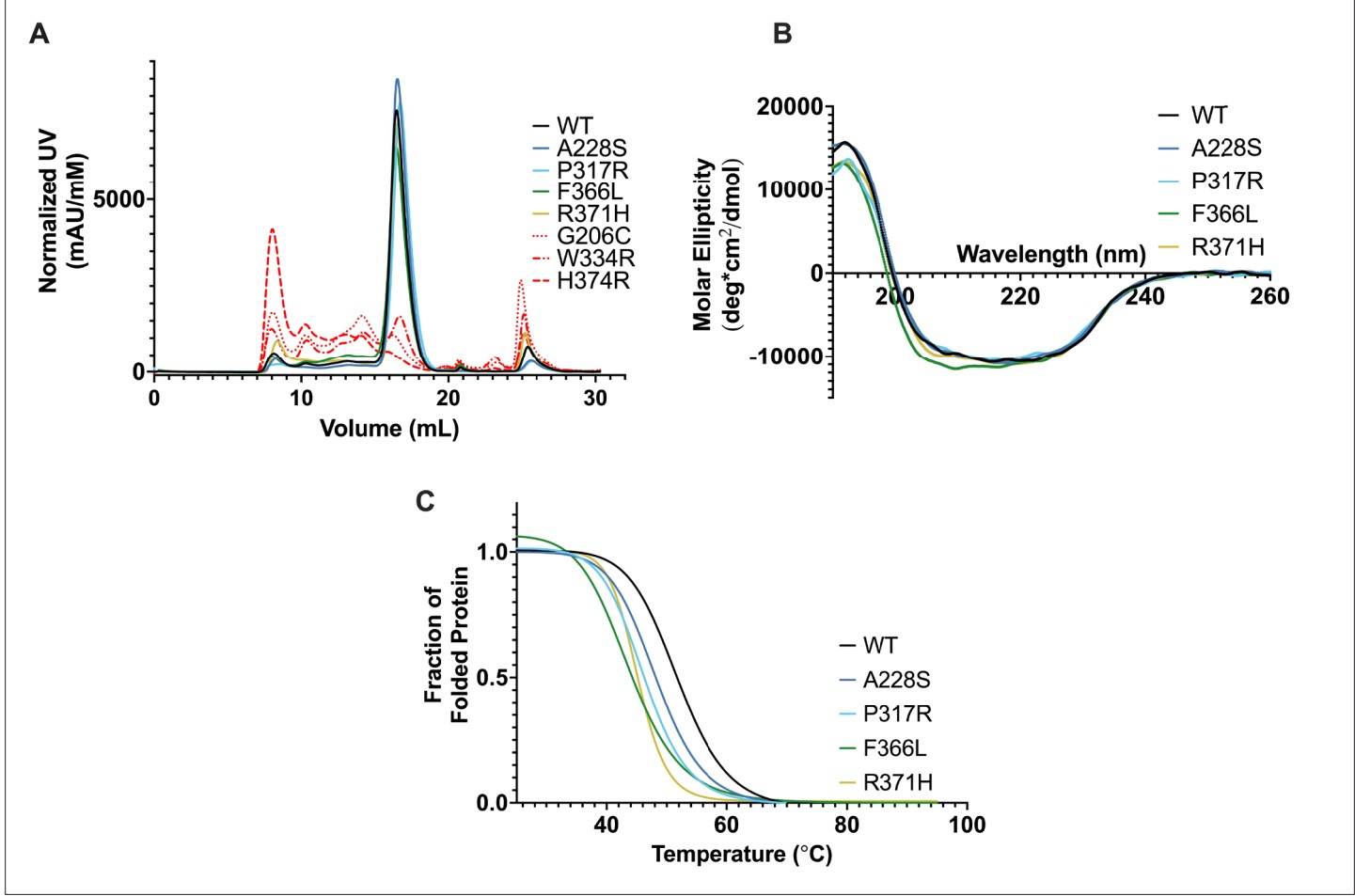

**Figure 3.** PHD2 mutants display instability through aggregation and decreased thermostability. (**A**) Chromatograms of PHD2 catalytic cores were acquired via size exclusion chromatography (SEC) on a Superdex200 column. Curves have been normalized according to molecular concentrations. Dashed red lines indicate mutants that were not purified (n = 2). (**B**) Circular dichroism (CD) was performed on PHD2 mutants to predict secondary structure variations. The far-UV spectra (190–260 nm) of the purified catalytic cores were measured and converted to molar ellipticity (n=3). (**C**) Molar ellipticity of PHD2 mutants was monitored at 220 nm from 25 to 95°C to evaluate thermal stability. Melting curves were generated from the CD melt (n=3). Molar ellipticity values were normalized and transformed into a fraction of folded protein and fitted with a sigmoidal curve. Melting temperatures ($T_m$) were determined using the EC50 of each curve. $T_m$ and $R_2$ are listed in *Table 1*.

The online version of this article includes the following source data and figure supplement(s) for figure 3:

**Figure supplement 1.** Purification of PHD2 181–426.

**Figure supplement 1—source data 1.** Unedited image of PHD2 purification SDS-PAGE gels stained with Coomassie blue corresponding to *Figure 3— figure supplement 1*.

**Figure supplement 1—source data 2.** Original image file corresponding to *Figure 3—figure supplement 1*.

**Table 1.** PHD2 mutant melting temperatures calculated via circular dichroism (CD).

|  | $T_m$ (°C) | $R^2$ |
| --- | --- | --- |
| WT | 51.9 | 0.871 |
| A228S | 48.3 | 0.707 |
| P317R | 46.2 | 0.748 |
| F366L | 43.8 | 0.878 |
| R371H | 45.1 | 0.843 |

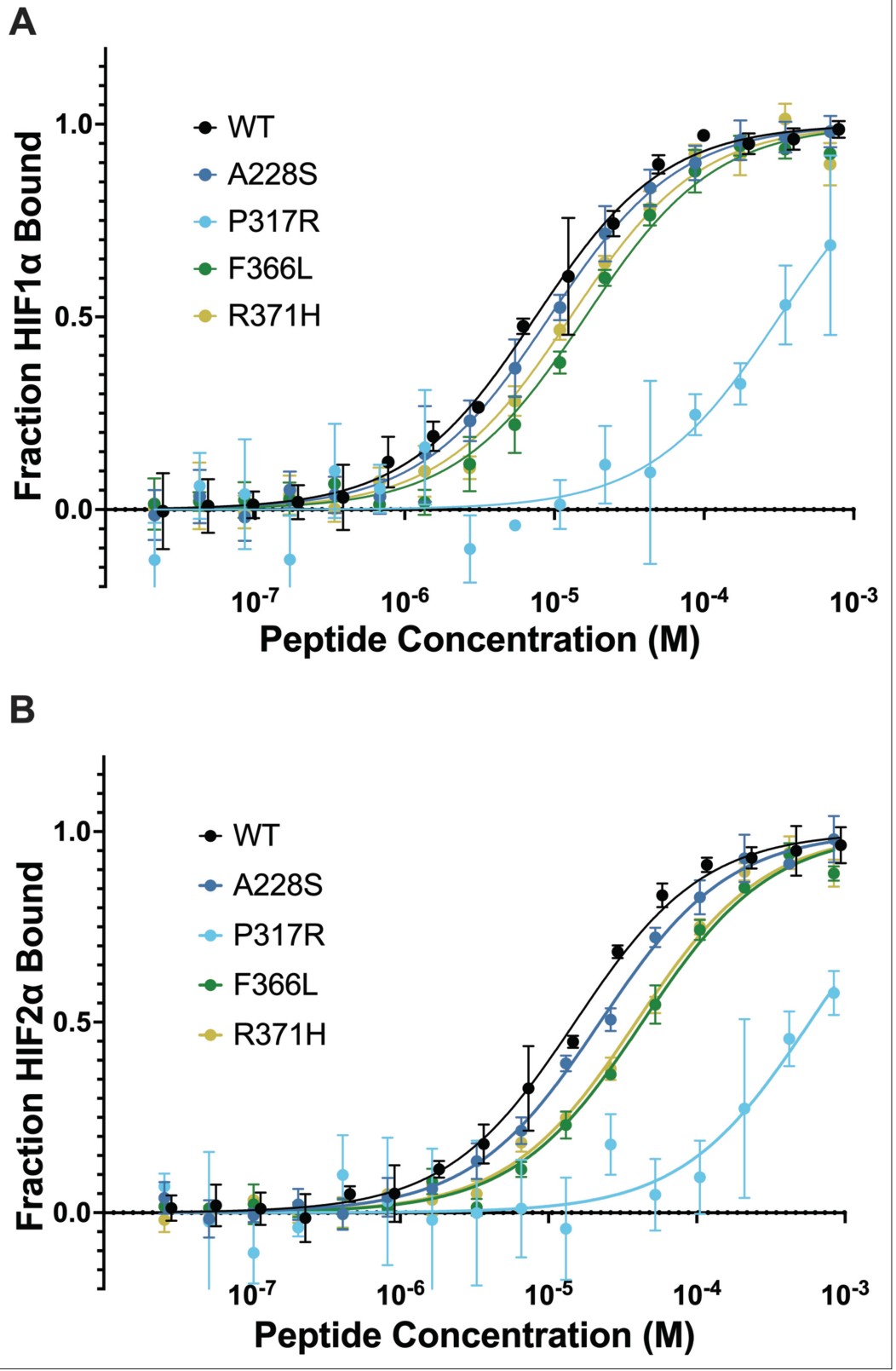

**Figure 4.** PHD2 mutants have minor binding defects to HIF1α peptides. Microscale thermophoresis was performed on fluorescently labeled PHD2 and HIF1α 555-574 CODD (**A**) or HIF2α 522-542 CODD (**B**) peptides. PHD2 P317R displayed a severe binding defect, whereas the other three mutants had minor binding defects. It is suspected that amine-reactive fluorescent labeling induced a binding defect on PHD2 P317R. Data points

*Figure 4 continued on next page*

*Figure 4 continued*

represent normalized mean value, and standard error is represented by error bars (n=3). K$_d$ values with standard deviation are listed in *Table 2*.

## PHD2 P317R binds effectively to both CODD and NODD via BLI

We performed bio-layer interferometry (BLI) using immobilized biotinylated HIF1α CODD and NODD peptides with purified PHD2 WT and P317R, which showed that P317R binding to the CODD peptide was only slightly weaker than PHD2 WT (*Table 3* and *Figure 5—figure supplement 2*). The discrepancy between the MST and BLI results may be due to the amine-linked fluorescent labeling of PHD2 P317R for MST, potentially disrupting binding to HIF1α CODD peptide. The removal of this label allowed for proper, unimpeded interaction to occur as observed with BLI. These results are consistent with the near-WT, efficient hydroxylation of HIF1α CODD by PHD2 P317R as revealed by NMR (see *Figure 5*).

The binding affinity between PHD2 P317R and HIF1αNODD was also measured to examine whether the catalytic defect observed via NMR would manifest as a binding defect. However, PHD2 P317R bound to NODD similarly to PHD2 WT, suggesting that its failure to hydroxylate P402 is unlikely attributable to binding. Binding between PHD2 WT and P317R with hydroxylated peptides was measured as a negative control, validating BLI as a viable technique for detecting binding defects (*Figure 5—figure supplement 3*).

## Discussion

Mutations occurring in the core components of the hypoxia sensing pathway have been shown to cause various phenotypes ranging from erythrocytosis and neuroendocrine tumors to hemangio-blastoma and renal cell carcinoma. Collectively, these diseases are referred to as pseudohypoxic diseases due to the inappropriate hypoxic response induced by the mutations. It is presumed that the abnormal response is the reason for overlapping phenotypes caused by mutations in different proteins of the same pathway (pVHL, HIF2α, and PHD2). Specifically, we have proposed that the common phenotypes of pseudohypoxic diseases are caused by the gain-of-function ability of HIFα, in particular HIF2α, to escape degradation in the presence of the disease-causing mutants (*Ohh et al., 2022*). It has been shown that VHL disease-causing mutants have varying ability to target HIF1/2α for ubiquitin-mediated destruction (*Rechsteiner et al., 2011*; *Knauth et al., 2009*), and Pacak-Zhuang syndrome-causing HIF2α mutants are able to escape recognition to a varying degree from both pVHL and PHD2 (*Tarade et al., 2018*; *Ferens et al., 2024*). It follows then that in the case of PHD2-driven erythrocytosis, PHD2 mutants would have a hindered ability to promote proper oxygen-dependent hydroxylation of HIF1α and -2α.

Complete deletion of PHD2 in mice causes embryonic lethality due to venous congestion and cardiomyopathy, theorized to be directly related to high blood volume induced upon PHD2 loss (*Minamishima et al., 2008*). Conditional inactivation of PHD2, whether globally (*Li et al., 2010*) or in renal EPO-producing cells (*Franke et al., 2013*), has been observed to cause erythrocytosis in mice. Moreover, while inactivation of PHD2 alone was sufficient to induce erythrocytosis, inactivating PHD1 or PHD3 alone did not result in erythrocytosis. However, when PHD1 or PHD3 was inactivated

**Table 2.** Microscale thermophoresis (MST) determined dissociation constants between PHD2 mutants and HIFα C-terminal oxygen-dependent degradation domain (CODD) peptides.

| PHD2 | K$_d$ (µM) | |
| --- | --- | --- |
| | HIF1α | HIF2α |
| WT | 7.5±0.52 | 15±1.1 |
| A228S | 9.3±0.57 | 22±1.6 |
| R371H | 13±1.6 | 39±4.4 |
| F366L | 16±1.8 | 43±4.7 |
| P317R | 320±20 | 580±38 |

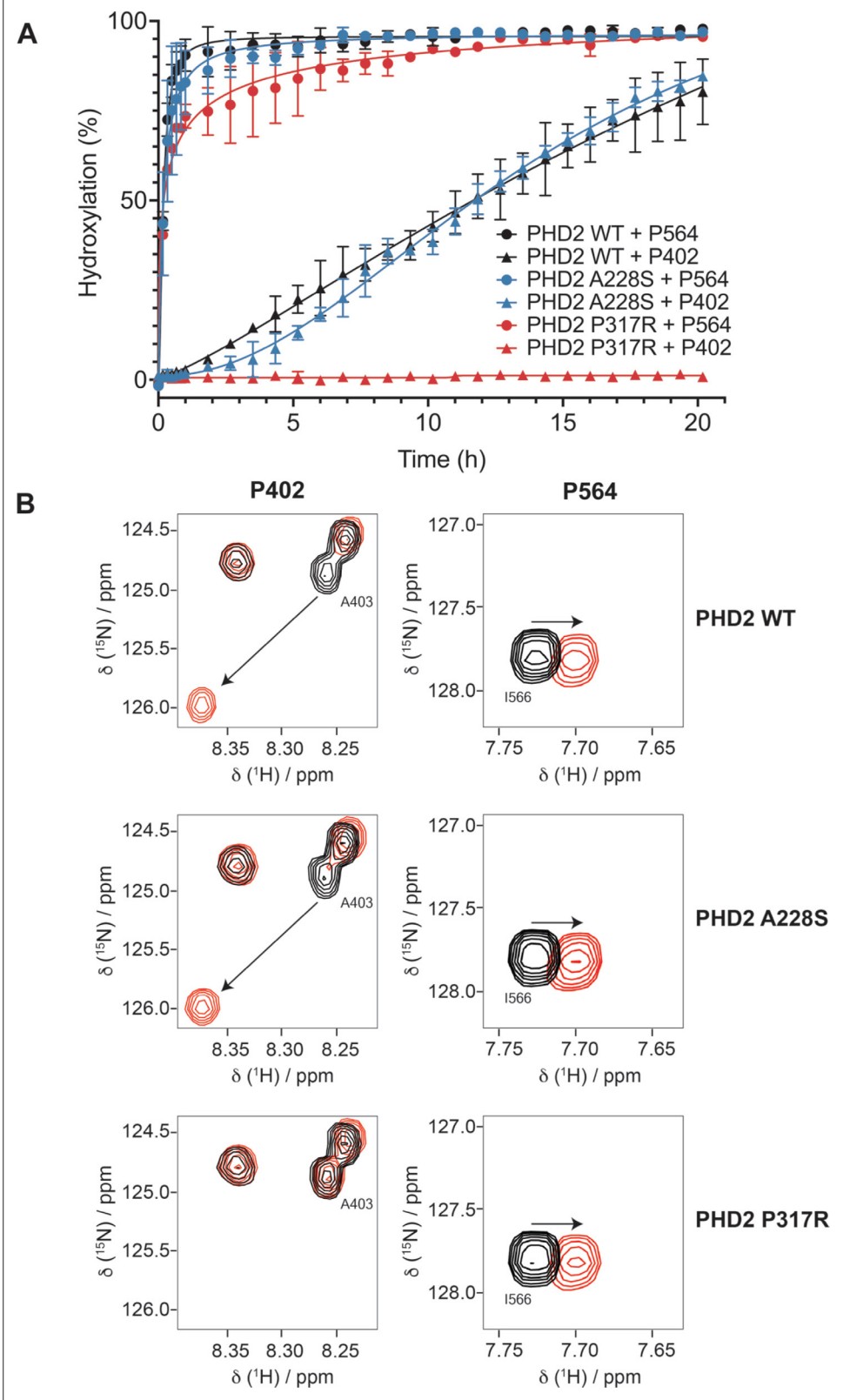

**Figure 5.** PHD2 P317R does not hydroxylate HIFα N-terminal oxygen-dependent degradation domain (NODD), while PHD2 A228S has very minor enzymatic defects. An assay measuring hydroxylation of HIF1α oxygen-dependent degradation domain (ODD) (394–574) by PHD2 via nuclear magnetic resonance (NMR) was performed. (**A**) Resonance shifting was monitored in real time to compare hydroxylation rates of A228S (blue), P317R (red), and

*Figure 5 continued*

wild-type (WT) (black) PHD2. PHD2 A228S displayed minorly impaired hydroxylation of both ODDs. PHD2 P317R displayed no activity on the P402 (NODD), while retaining near WT activity on P564 (C-terminal ODD [CODD]). Data points represent mean value, and standard deviation is represented by error bars (n=2). (**B**) HSQC spectra display the resonance shifting pattern of HIF1a ODD upon prolyl-hydroxylation catalyzed by PHD2 (WT, P317R, A228S) over the course of 20.2 hr. Neighboring residues, A403 and I566, were used to monitor hydroxylation of P402 and P564, respectively. Spectra recorded at 0 hr are shown in black while spectra recorded at the endpoint (20.2 hr) are shown in red.

The online version of this article includes the following figure supplement(s) for figure 5:

**Figure supplement 1.** Tracking proline hydroxylation using CON versus HSQC nuclear magnetic resonance (NMR).

**Figure supplement 2.** Bio-layer interferometry (BLI) shows PHD2 P317R does not have a severe binding defect for HIF1α C-terminal oxygen-dependent degradation domain (CODD) or HIF1α N-terminal oxygen-dependent degradation domain (NODD).

**Figure supplement 3.** Validation of bio-layer interferometry (BLI) using hydroxylated peptides as a negative control.

in addition to PHD2, erythrocytosis phenotype worsened, suggesting a supporting role for the PHD paralogs (*Takeda et al., 2008*).

Here, we examined seven disease-causing PHD2 mutations across the catalytic core, the most frequently mutated region within PHD2. G206C, W334R, and H374R exhibited gross protein stability defects and were prone to severe aggregation, which precluded their purification necessary for further biophysical analyses. One of these mutants, H374R, is located at the catalytic site integral to the function of PHD2, which combined with the stability defect likely abrogated its ability to promote the oxygen-dependent hydroxylation of ectopic HIF1α, leading to increased HRE-luciferase transcriptional activity in *PHD2-/-* HEK293A cells. Previous work has shown that mutating the analogous His iron-coordinating residues to a charged residue (Arg or Glu) in the prolyl-4-hydroxylase responsible for hydroxylating prolines of collagen abolishes enzymatic activity (*Myllyharju and Kivirikko, 1997*). Given the high structural conservation in the Fe/2OG-oxygenase enzyme family (*Clifton et al., 2006*), it can be predicted that a comparable mutation on PHD2 would also abolish activity. Structurally, G206 is located at a sharp turn in PHD2, requiring the flexibility afforded by glycine. This property may be difficult to substitute without detriment as is observed with G206C. W334 is in a loop that connects the beta strands, making up the highly conserved jelly roll structure of 2OG oxygenases (*Clifton et al., 2006*). A mutation at W334 may impact the orientation of this loop and destabilize the structure of PHD2. Our data corroborates this theory by revealing W334R to be highly unstable in cells and in vitro. However, PHD2 G206C and W334R mutants showed similar capacity as the PHD2 WT to inhibit the transcriptional activity of ectopic HIF1α. This may have been due to the nature of transfected PHD2 in which mutant PHD2s that are prone to aggregation and instability are being translated throughout the luciferase assay, which may generate a small pool of monomeric PHDs that is sufficient to inhibit the transcriptional activity of HIF1α comparable to PHD2 WT.

The remaining four mutants (A228S, P317R, F366L, and R371H) not prone to aggregation displayed an overall structure similar to PHD2 WT, as determined by CD, and with the exception of PHD2 P317R (discussed later), they were able to effectively suppress the ectopic HIF1α transcriptional activity. These mutants also exhibited slight thermal instability with mildly weaker binding affinity to HIF1α CODD peptide, which may be sufficient to cause the disease phenotype in vivo. PHD2 A228S exhibited the

**Table 3.** Bio-layer interferometry (BLI) determined binding constants between PHD2 wild-type (WT) and P317R and HIF1α N-terminal oxygen-dependent degradation domain (NODD) and C-terminal oxygen-dependent degradation domain (CODD) peptides.

|  | $K_d$ (µM) | $k_a$ (×10⁴) (1/Ms) | $k_d$ (×10⁻³) (1/s) |
|---|---|---|---|
| PHD2 WT+HIF1α CODD | 2.1±0.8 | 5.2±1.4 | 101±20 |
| PHD2 P317R+HIF1α CODD | 6.0±0.7 | 0.21±0.11 | 12.3±5 |
| PHD2 WT+HIF1α NODD | 2.0±0.8 | 0.26±0.093 | 4.7±0.2 |
| PHD2 P317R+HIF1α NODD | 2.0±1.0 | 0.17±0.049 | 3.0±0.7 |

most minute of defects among all mutants tested and concordantly showed marginal reduction in enzymatic activity as measured via time-resolved NMR. Notably, despite these defects, albeit subtle, it is evident that the HRE-luciferase reporter assay is in general not reliable to reveal the disease-causing mutations on the functional output of PHD2 mutants, including those prone to aggregation (G206C and W334R). This represents a limitation in the interpretation of the HRE-luciferase reporter assay as mutants with clear structural defects do not necessarily impact downstream hypoxic signaling.

We recently developed a time-resolved NMR approach to directly and simultaneously measure PHD2-mediated hydroxylation of P564 within CODD and P402 within NODD in the context of full-length HIF1α ODD. We showed that P564 is the predominant oxygen-dependent hydroxylation site over P402. Considering that pVHL only requires one hydroxylated proline for binding and triggering rapid ubiquitin-mediated proteasomal degradation of HIF1α (*He et al., 2021*; *Chan et al., 2005*; *He et al., 2024*; *Tian et al., 2011*), the biological utility of P402 was thought to be inconsequential in normal hypoxic response. Here, our results revealed a previously unappreciated intricacy in the enzymatic selectivity of PHD2 P317R between CODD and NODD, challenging the current notion of NODD insignificance. NMR results showed that the disease-causing PHD2 P317R retained near-WT activity on CODD P564 but had negligible activity on NODD P402. Furthermore, despite effective activity on P564, ectopic PHD2 P317R generated abnormally high HRE-driven luciferase reporter signal, suggesting the negative impact of defective P402 hydroxylation on an otherwise rapid oxygen-dependent degradation of HIF1α. This is congruous with Schofield's earlier work that showed, using an indirect metabolic method with individual peptides, no activity on NODD by PHD2 P317R while effectively hydroxylating CODD (*Chowdhury et al., 2016*). Given the pathogenic nature of the P317R mutation, it appears likely that this selective defect on HIF1α NODD contributes to the development of erythrocytosis, and, therefore, the involvement of NODD in normal oxygen-sensing pathway is not insignificant.

The molecular basis for PHD2 P317R's selective defect for NODD is currently unknown. Comparison of WT and P317R PHD2 crystal structures suggests that P317 makes hydrophobic contacts with the LXXLAP motif on HIFα, and R317 may interact differently with this motif, impacting HIFα binding (*Chowdhury et al., 2016*). Intriguingly, BLI results showed that PHD2 P317R binds to both CODD and NODD peptides comparable to PHD2 WT. Substrates of Fe/2OG-oxygenases typically bind the enzymes non-covalently with a prime separation distance of 4–5 Å between the target atom and the catalytic iron (*Aik et al., 2012*). One possibility is that the Pro to Arg substitution at position 317 may increase the distance between HIF1α P402 and the active site within the catalytic core without compromising the interaction between the substrate and enzyme. However, such mutation did not markedly impact the efficient turnover of P564, suggesting that the varied residues between NODD and CODD may influence the substrate selectivity of PHD2.

Some mutants, namely A228S and F366L, appear to be like PHD2 WT according to the tests performed. Minor binding and stability defects were observed with both mutants, yet it remains unclear if this is the pathogenetic mechanism. Upon structural review, it becomes clear that F366L would disrupt the hydrophobic core of PHD2, and this instability can be observed in the CHX assay but not in the other assays. An A228S-caused defect is less obvious. A228 is located on the surface of the catalytic core, and while it is possible a residue change to polar serine could impact how PHD2 interacts in solution, this remains to be validated. This raises the notion that these mutants may have defects interacting with previously proposed, though controversial, alternate non-HIF substrates (*Cockman et al., 2019*).

PHD2-driven erythrocytosis is a relatively newly discovered genetic disease with the first reported case in 2006. Since then, there has been a large increase in reports owing to the addition of *EGLN1* mutant screening in recent years for cases of idiopathic erythrocytosis (*Rechsteiner et al., 2011*). As further awareness of the disease grows, screening will increase, leading to the likely identification of more disease-associated mutations. This study, therefore, will add to the groundwork knowledge of how mutant PHD2 induces disease in the context of the hypoxic response.

## Materials and methods
### Plasmids, antibodies, and peptides
Previously described plasmids were used as follows: pET-46-HIS$_6$-PHD2 (*Tarade et al., 2018*), pGL3-VEGFA-HRE (*Ferens et al., 2024*), pCDF-Ek/LIC-StrepII-HIS$_6$-HIF1α ODD (*He et al., 2024*), pcDNA3-HA-HIF1α (*He et al., 2021*). pRL-SV40 was procured from Promega. pcDNA3-3XFLAG-EGLN1 was generated by amplifying the EGLN1 gene from HA-EglN1-pcDNA3 (a gift from William Kaelin; Addgene plasmid #18963) using primers 5' – GCG GCG GGA TCC ATG GCC AAT GAC AGC G – 3' and 5' – GCG GCG TCT AGA CTA GAA GAC GTC TTT ACC GAC – 3'. The product was digested with BamHI/XbaI and subcloned into pcDNA3 vector containing an N-terminal 3XFLAG tag. pX330-U6-Chimeric_CBh-hSpCas9 was gifted to the lab from Feng Zhang (Addgene plasmid #42230). PHD2 mutants of pET-46-HIS$_6$-PHD2 and pcDNA3-3XFLAG-PHD2 were generated through QuikChange site-directed mutagenesis (Agilent). Mutant sequences were confirmed via DNA sequencing. The following antibodies were used: α-HA (C29F4) from Cell Signaling, α-FLAG M2 (F1804) from Sigma-Aldrich, α-Vinculin (V9264) from Sigma-Aldrich, α-HIF1α (36169) from Cell Signaling, and α-PHD2/EGLN1 (D31E11) from Cell Signaling. Primary antibodies were diluted in TBS-T (20 mM Tris pH 7.6, 137 mM NaCl, 0.05% [vol/vol] Tween 20) with 0.02% (wt/vol) sodium azide for immunoblotting. Peptides were N-terminally acetylated for MST and biotinylated for BLI. All peptides were amidated at the C-terminus. For MST, peptides were reconstituted in 50 mM Tris (tris(hydroxymethyl)aminomethane) pH 8.0 buffer. For BLI, peptides were reconstituted in 100% (vol/vol) DMSO.

### PHD2 -/- HEK293A cells
WT human epithelial kidney (HEK293A) cells acquired from the American Type Culture Collection (CRL-1573) were subjected to the clustered regularly interspaced short palindromic repeats (CRISPR) method to generate homozygous PHD2 knockout (-/-) cells. The CRISPOR design tool was used to generate CRISPR guide RNAs (gRNAs) targeting exon 1 in *EGLN1* (*Concordet and Haeussler, 2018*). Forward and reverse gRNA oligonucleotides were phosphorylated and annealed before ligation into BbsI-digested pX330-U6-Chimeric_CBh-hSpCas9. WT HEK293A cells were transfected with pX330-EGLN1 gRNA using Lipofectamine 2000 according to the manufacturer's instructions. Single-cell colonies were seeded into 96-well plates from the previously transfected cells at a density of 1.25 cells/ml. 200 µl of cells were seeded into each well, resulting in a density of ~0.25 cells/well. Single-cell clones were screened for PHD2 knockout via immunoblotting for PHD2. Western blots validating PHD2 knockout are shown in *Figure 2—figure supplement 1*.

### Dual luciferase assay
A dual luciferase assay was performed according to a modified previously reported protocol (*Ferens et al., 2024*). PHD2 -/- HEK293A cells were grown and maintained in Dulbecco's Modified Eagle Medium (DMEM, Wisent) with 10% (vol/vol) Fetal Bovine Serum (Wisent) and 100 IU/ml penicillin and streptomycin in a humidified incubator at 37°C and 5% $CO_2$. Transfection complexes consisted of 2 µg total DNA: 0.4 µg pcDNA3-HA, 0.9 µg pGL3-VEGFa-HRE, 0.1 µg pRL-SV40, 0.4 µg pcDNA3-HA-HIF1α, and 0.2 µg pcDNA3-FLAG3X-PHD2 (WT or mutant). Complexes were incubated for 15 min at room temperature with 8 µg polyethyleneimine pH 7.2 (Polysciences) in 400 ml of OptiMEM Reduced Serum Media (Gibco). Following incubation, transfection complexes were added to 5×10$^5$ PHD2-/- HEK293A cells suspended in 400 µl OptiMEM and incubated for 5 min. Cells were transferred to six-well plates and incubated at 37°C, 5% $CO_2$ for 24 hr. Cells were lysed and processed according to the Promega Dual-Luciferase Reporter Assay System (#E1960) instructions. Firefly luciferase and Renilla luciferase activity were measured on a Varioskan Lux microplate reader (Thermo Fisher Scientific). Luciferase activity was normalized by dividing Firefly RLU values by the constitutively expressed control Renilla RLU values. Normalized RLU values were transformed by setting the RLU value of the WT PHD2 sample to 1 and dividing the other sample values by the same transformation factor. Three biological replicates were completed for each mutant. A two-tailed t-test was performed to determine statistically significant differences between samples. A p-value below 0.0332 was considered significant. Protein levels of PHD2 were determined by immunoblotting for FLAG3X-PHD2. Vinculin was probed as a loading control.

## CHX chase assay

HEK293A cells were cultured as described above. Transfection complexes consisting of 1.5 µg total DNA were incubated with 6 µg polyethyleneimine pH 7.2 in 400 µl of OptiMEM for 15 min at room temperature. Transfection complexes consisted of either 0.5 µg (WT, A228S, P317R, F366L, R371H) or 1.5 µg (G206C, W334R, H374R) of pcDNA3-FLAG3X-PHD2. Additional pcDNA3-FLAG3X-PHD2 was transfected for less stable mutants to ensure equal amounts of PHD2 expression. The total amount of DNA was brought to 1.5 µg with the addition of 1 µg of pcDNA3 if necessary. Transfection complexes were added to $1.5 \times 10^5$ cells suspended in 400 µl OptiMEM and incubated for 5 min. Cells were plated in 60 mm plates with 5 plates for each mutant and 2 for WT. Cells were incubated for 24 hr at 37°C and 5% $CO_2$. Following incubation, media was replaced to remove polyethyleneimine, and fresh media with 10 µg/ml CHX was added. Cells were incubated in CHX-supplemented media for 0, 3, 6, 12, and 24 hr. WT samples were incubated only for 0 and 24 hr. Cells were harvested through manual scraping and lysed via sonication in 250 µl of cold EBC lysis buffer (50 mM Tris, 120 mM NaCl, 0.5% [vol/vol] NP-40) supplemented with 1X protease inhibitor cocktail (BioShop PIC002.1). Protein levels were determined via Bradford assay, and equal protein amounts of lysate were denatured by boiling for 5 min with 3X sample buffer (62.5 mM Tris, 10% [vol/vol] glycerol, 2% [wt/vol] SDS, 0.01% [wt/vol] bromophenol blue). Samples were then resolved on a 10% acrylamide SDS-PAGE gel for 90 min at 120 V. Proteins were transferred to polyvinylidene fluoride membrane via wet electrophoretic transfer for 70 min at 110 V. Membranes were blocked for 60 min at room temperature in 5% (wt/vol) skim milk in TBS-T. Membranes were then incubated with α-FLAG-M2 (1:1500) and α-vinculin (1:1500) at 4°C overnight. Following three washes with TBS-T, membranes were incubated with HRP-conjugated secondary antibodies (1:10,000 in 5% [vol/vol] skim milk in TBS-T) for 60 min at room temperature. Proteins were detected using a chemiluminescence solution and imaged with a ChemiDoc MP Imaging System (Bio-Rad). Immunoblot band density was quantified using Bio-Rad Image Lab densitometry software. FLAG3X-PHD2 levels were normalized according to corresponding vinculin levels. Each 0 hr sample was set to 1, and each other time point was divided by this value to reflect relative protein levels. Three biological replicates were completed for each mutant. A two-tailed t-test was performed to identify significant differences between 0 and 24 hr timepoints. A p-value below 0.0332 was considered significant.

## PHD2 (181–426) expression and purification

BL21 (DE3) *E. coli* cells (Novagen, #69450) were transformed with pET-46-HIS$_6$-PHD2 (WT or mutant). 1 l Luria Broth (LB) cell cultures were grown at 37°C until $OD_{600}$ reached ~0.8. Cultures were then induced with a final concentration of 1 mM isopropyl β-D-1-thiogalactopyranoside for 16 hr at 16°C. Cultures were harvested via centrifugation and stored at –80°C for future use. His$_6$-PHD2 (181–426) was purified according to a previously described protocol (*Hewitson et al., 2007*). Bacterial pellets were resuspended in lysis buffer (50 mM Tris-HCl pH 7.9, 500 mM NaCl, 5 mM imidazole) supplemented with 1X SigmaFast protease inhibitor cocktail (Sigma-Aldrich). Cells were lysed with 3 passes through a cell disruptor at 30 kpsi. Cell lysate was cleared via centrifugation at $30,000 \times g$ for 45 min. Cleared lysates were applied to a 2 ml Ni-NTA resin (Thermo Fisher). The resin was washed with 10 ml of lysis buffer followed by 2 separate 30 ml washes with Wash buffer (50 mM Tris-HCl pH 7.9, 500 mM NaCl, 30 mM imidazole). His$_6$-PHD2 was eluted from the resin with 10 ml of elution buffer (50 mM Tris-HCl pH 7.9, 500 mM NaCl, 1 M imidazole). The eluate was dialyzed against 2 l of dialysis buffer (50 mM Tris-HCl pH 7.5) using a 10 kDa cutoff dialysis membrane. After 4 hr, the dialysis buffer was replaced with 2 fresh liters, and 1.5 units of thrombin was added per mg of His$_6$-PHD2. His$_6$-PHD2 was dialyzed overnight with thrombin at 4°C. The dialyzed PHD2 was applied to Ni-NTA resin equilibrated with dialysis buffer to separate thrombin-cleaved protein from uncleaved protein and His$_6$ tags. The resin was washed with dialysis buffer supplemented with 5 mM imidazole. The flowthrough and wash were combined and concentrated to ~600 µl using an Amicon 10 kDa cutoff centrifugal concentrator (Millipore). The concentrated PHD2 was cleared at 13,000 rpm for 10 min then applied to a Superdex 200 Increase 10/300 (Cytiva) column equilibrated with dialysis buffer or labeling buffer (50 mM HEPES [*N*-2-hydroxyethylpiperazine-*N*'-2-ethanesulfonic acid] pH 7.5, 150 mM NaCl). Elution was monitored via UV absorbance, and samples with PHD2 were collected. SDS-PAGE gels were run to confirm purity (*Figure 3—figure supplement 1*). PHD2 proteins were concentrated to >2 mg/ml. The final concentration was measured via UV absorbance at 280 nm ($\varepsilon_{0.1\%} = 1.34$). Unlabeled protein was flash-frozen

and stored at –80°C for use in CD and NMR. For size exclusion column analysis, the protein was concentrated to around 2 mg/ml prior to application to the Superdex 200 Increase 10/300 (Cytiva) column and the His6 tag was not cleaved.

PHD2 proteins used for MST were labeled with Alexa Fluor-647 *N*-hydroxylsuccinimide ester dye (Thermo Scientific) following purification according to a previously reported protocol (*Ferens et al., 2024*). PHD2 proteins were diluted to 1 mg/ml and Alexa Fluor-647 dye was added to 500 µl of protein in a 4:1 molar ratio (dye: PHD2). Following incubation at room temperature in the dark for 1 hr, excess dye was removed using a PD-10 Sephadex G-25 M desalting column (Cytiva) equilibrated with dialysis buffer. Labeled protein was subjected to a 10-fold dilution in dialysis buffer and was then concentrated using a 10 kDa cutoff Amicon centrifugal concentrator to remove excess dye. Alexa Fluor-647-labeled PHD2 was then brought to 10 mM in dialysis buffer, flash-frozen, and stored at –80°C for use in MST.

## HIF1α ODD (394–574) expression and purification

DNA corresponding to the sequence of human HIF-1α ODD (residues 394–574) with a C-terminal Strep Tag II (WSHPQFEK) was cloned into a pCDF Ek/LIC vector (EMD Millipore). Thrombin cleavage sites were included after the N-terminal-His$_6$ tag and before the C-terminal Strep Tag II. HIF-1α ODD was expressed in Rosetta-2 (DE3) cells. To produce $^{15}$N-labeled proteins, cells were grown in 2X M9 minimal media, supplemented with a final concentration of 1 g/l $^{15}$NH$_4$Cl (ACP Chemicals) and 4 g/l D-glucose (Bioshop). Expression and purification of HIF-1α ODD was previously described (*He et al., 2024*). Briefly, cleared lysate was applied to Ni-NTA resin (Thermo Fisher) and eluted with 300 mM imidazole. The elution was then incubated with StrepTactin XT 4Flow resin (IBA Lifesciences) over-night. Protein was eluted from the StrepTactin resin with 50 mM biotin. This process was repeated with the flowthrough from the StrepTactin beads to capture remaining unbound protein. Protein was concentrated and run on a Bio-Rad SEC650 10/300 column. Fractions containing HIF-1α ODD were then run on a HiResQ 5/50 (Cytiva) anion exchange column. After tag cleavage with thrombin, purified HIF-1α ODD was concentrated to 0.9 mM and stored at –80°C for future use.

## Circular dichroism

Purified PHD2 proteins were buffer exchanged on PD-10 Sephadex G-25 M desalting columns (Cytiva) for 5 mM KH$_2$PO$_4$/K$_2$HPO$_4$ pH 7.4 buffer. Proteins were then diluted to 0.1 mg/ml. Far-UV wavelength scans were collected from 190 to 260 nm on a Jasco-J-1500 CD spectrophotometer with a 1 mm pathlength quartz cuvette (Helma). CD values were averaged from 20 accumulations, baseline buffer subtracted, and converted into mean residue ellipticity (degrees cm$^2$/dmol). Spectra were smoothed using Jasco Spectra Analysis software. Wavelength scans were run in triplicate, and the average mean residue ellipticity was plotted with other PHD2 CD curves.

Melting curves were generated by monitoring CD values for each PHD2 protein from 25°C to 95°C at 220 nm with 1°C steps. Values were baseline buffer subtracted and converted to mean residue ellipticity. Triplicate non-transformed data was plotted. Curves were transformed by setting the average peak value between 25°C and 37°C to 1.0 and normalizing other points according to the same normalization factor. Curves were fitted with a sigmoidal curve on Prism. Corresponding R$^2$ values are recorded in *Table 1*. Melting temperature (T$_m$) for each protein was determined from the EC50 of each sigmoidal curve. Curves were plotted using GraphPad Prism version 10.2.3.

## Microscale thermophoresis

MST binding tests and analysis were performed according to a previously recorded protocol (*Ferens et al., 2024*). A Monolith NT.115 (Nanotemper Technologies) was used at room temperature to record MST measurements. 2X stock solutions of Alexa Fluor-647-labeled PHD2 with necessary hydroxylation reagents were made (50 nM Alexa Fluor-647-PHD2, 50 mM Tris-HCl pH 7.5, 1 mM N-oxalylglycine (NOG), 10 µM ferrous sulfate, 5 mg/ml BSA). ~1 mg lyophilized HIF1α $_{555-574}$ CODD and HIF2α $_{522-542}$ CODD peptides were dissolved in 200 µl of 50 mM Tris-HCl pH 8.0, and their concentrations were measured via UV absorbance with the following extinction coefficient: 1490 M$^{-1}$ cm$^{-1}$. Starting with the stock HIFα peptide solutions, twofold serial dilutions were performed to generate 16 different 15 µl HIFα concentrated solutions. 15 µl of 2X Alexa-647-PHD2 solution was then added to each HIFα peptide dilution resulting in a final Alexa-647-PHD2 concentration of

25 nM. The solutions were incubated at room temperature for 10 min, then loaded into standard MST capillaries (Nanotemper Technologies). MST measurements were recorded using 20% fluorescence excitation LED power and 40% (medium) infrared (IR) laser power. Measurements were recorded for 5 s prior to IR laser activation, 20 s during IR activation, and 3 s following IR deactivation. Technical triplicates were analyzed and fit with a 1:1 $K_d$ binding model using M.O. Affinity Analysis software (Nanotemper Technologies). Curves were plotted on GraphPad Prism (version 10.2.3).

## Bio-layer interferometry

The binding affinities of HIF1α peptides and WT or P317R PHD2 were measured by BLI using a single-channel BLItz instrument (Sartorius). Purified PHD2 was concentrated to 0.45 mM and diluted into BLI Kinetic Buffer (20 mM Tris-HCl pH 7.4, 100 mM NaCl, 1 mM L-ascorbic acid [Sigma], 100 μM ferrous chloride tetrahydrate [Sigma], 1 mM NOG [Sigma], 0.1% [wt/vol] Bovine Serum Albumin, and 0.1% [vol/vol] Tween-20). Following reconstitution to a final concentration of 2 mg/ml in DMSO, biotinylated HIF1α peptides were diluted 200-fold in the BLI Kinetic Buffer. HIF1α peptides were immobilized on a streptavidin-coated biosensor (Sartorius) for 120 s before measuring association to WT PHD2 or P317R PHD2 in a concentration series over 120 s. Subsequently, the biosensor was immersed in the BLI Kinetic Buffer for 120 s to measure dissociation of the analyte. The sensorgrams were referenced and step corrected, and the binding affinities and kinetic rates were calculated based on global fit of the data to a 1:1 binding model using the BLItz Pro v.1.3.0.5 software. All binding measurements were performed in technical triplicates.

## Real-time prolyl-hydroxylation kinetic measurement

Real-time prolyl-hydroxylation of HIF1α ODD was measured according to a previously reported protocol based on NMR (*He et al., 2024*). NMR experiments were performed at the UHN Nuclear Magnetic Resonance Core Facility on a Bruker 800 MHz US2 equipped with a NEO console and a 5 mm triple-resonance $^{13}$C optimized TXO cryoprobe, Z-gradient. NMR samples were prepared in 50 mM Tris-HCl pH 7.5, 100 mM NaCl, and 0.5 mM Tris(2-carboxyethyl) phosphine hydrochloride (TCEP-HCl) (NMR buffer) supplemented with 5% $D_2O$. Prior to the experiments, NMR buffer was gassed with 99.6% oxygen (Messer) for 40 min and then used to dilute 0.9 mM of HIF1α ODD to a final concentration of 0.18 mM. Samples were transferred to a 5 mm NMR tube pre-purged with 99.6% oxygen for 10 min. The real-time hydroxylation experiments were conducted at 25°C using the $^1$H-$^{15}$N HSQC pulse sequence with sodium 2,2-dimethyl-2-silapentane-5-sulfonate as reference. For each experiment, an $^1$H-$^{15}$N HSQC spectrum of HIF1α ODD was acquired prior to initiating the hydroxylation reaction with the addition of purified WT or mutant PHD2 at a molar ratio of 1:20 versus HIF1α ODD with 5 mM α-ketoglutaric acid disodium salt dihydrate (Sigma), 2 mM L-ascorbic acid (Sigma), and 0.1 mM ferrous chloride tetrahydrate. Multiple $^1$H-$^{15}$N HSQC spectra with acquisition time of 602 s were acquired over the course of 20.18 hr. After six consecutive acquisitions of $^1$H-$^{15}$N HSQC spectra, the subsequent spectra were acquired with interscan delays of 2400 s. Kinetic measurements with WT, A228S, and P317R PHD2 were performed in technical duplicates.

Real-time hydroxylation of P402 and P564 of HIF1α was measured by monitoring resonance shifting of A403 and I566 (*He et al., 2024*), respectively, in all $^1$H-$^{15}$N HSQC spectra. Both resonances shifted as reflections of changes in the local chemical environments upon prolyl-hydroxylation. Reaction progress was reported as the peak intensity of the shifted resonance over the sum of peak intensities for both shifted and unshifted resonances. All spectra were processed with the NMRPipe software (*Delaglio et al., 1995*) and analyzed using the CcpNmr Version-3 software (*Skinner et al., 2016*). Kinetic curves were plotted using GraphPad Prism (version 10.2.3).

## Acknowledgements

We thank the members of the Lee and Ohh labs for their critical comments and helpful discussions. This study was supported by funds from the Canadian Institutes of Health Research (MOP-133694 to JEL and PJT-191811 to MO) and the generous donations from Colorworks Canada (to MO). The UHN NMR Core Facility was supported by the Princess Margaret Cancer Foundation. Infrastructure for biophysics and NMR was funded by the Canada Foundation for Innovation (CFI).

# Additional information

### Funding

| Funder | Grant reference number | Author |
|---|---|---|
| Canadian Institutes of Health Research | PJT-191811 | Michael Ohh |
| Canadian Institutes of Health Research | MOP-133694 | Jeffrey E Lee |

The funders had no role in study design, data collection and interpretation, or the decision to submit the work for publication.

### Author contributions

Cassandra C Taber, Conceptualization, Data curation, Formal analysis, Validation, Investigation, Visualization, Methodology, Writing – original draft, Writing – review and editing; Wenguang He, Formal analysis, Investigation, Methodology, Writing – original draft, Writing – review and editing; Geneviève MC Gasmi-Seabrook, Resources, Software, Formal analysis, Methodology, Writing – review and editing; Mia Hubert, Investigation, Writing – review and editing; Fraser G Ferens, Conceptualization, Writing – review and editing; Mitsuhiko Ikura, Resources, Formal analysis, Supervision, Writing – review and editing; Jeffrey E Lee, Resources, Supervision, Funding acquisition, Writing – review and editing; Michael Ohh, Conceptualization, Resources, Supervision, Funding acquisition, Writing – original draft, Project administration, Writing – review and editing

### Author ORCIDs

Michael Ohh ⬚ https://orcid.org/0000-0001-7600-4751

Reviewer #1 (Public review): https://doi.org/10.7554/eLife.107121.3.sa1
Reviewer #2 (Public review): https://doi.org/10.7554/eLife.107121.3.sa2
Reviewer #3 (Public review): https://doi.org/10.7554/eLife.107121.3.sa3
Author response https://doi.org/10.7554/eLife.107121.3.sa4

# Additional files

### Supplementary files

Source data 1. Raw and transformed data for *Figures 2–5*.

Supplementary file 1. Clinical case reports of disease-causing PHD2 mutations.

Supplementary file 2. HIFα peptide sequences.

MDAR checklist

### Data availability

The authors declare that the clinical and numerical data supporting the findings of this study are available in *Supplementary file 1* and *Source data 1*, respectively. Unedited images are provided for immunoblot membranes and SDS-PAGE gels as source data files alongside the corresponding figures.

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
