## [Editor Report · eLife Assessment]

In this **valuable** study, Taber et al. used a battery of biophysical and structural approaches to characterize the impact of erythrocytosis-related mutations in prolyl hydroxylase domain protein 2 (PHD2). The authors show that PHD2 mutant proteins are destabilized, thus supporting the tenet that dysregulation of PHD2/hypoxia induced factor (HIF) axis underpins erythrocytosis, while providing **solid** evidence that N-terminal ODD prolyl hydroxylation of HIF is indispensable for these phenotypes. These findings were found to be of interest for researchers focusing on oxygen sensing in homeostasis and pathological states.

---

## [Referee Report · Reviewer #1 (Public review)]

Summary:

Taber et al report the biochemical characterization of 7 mutations in PHD2 that induce erythrocytosis. Their goal is to provide a mechanism for how these mutations cause the disease. PHD2 hydroxylates HIF1a in the presence of oxygen at two distinct proline residues (P564 and P402) in the "oxygen degradation domain" (ODD). This leads to the ubiquitylation of HIF1a by the VHL E3 ligase and its subsequent degradation. Multiple mutations have been reported in the EGLN1 gene (coding for PHD2), which are associated with pseudohypoxic diseases that include erythrocytosis. Furthermore, 3 mutations in PHD2 also cause pheochromocytoma and paraganglioma (PPGL), a neuroendocrine tumour. These mutations likely cause elevated levels of HIF1a, but their mechanisms are unclear. Here, the authors analyze mutations from 152 case reports and map them on the crystal structure. They then focus on 7 mutations, which they clone in a plasmid and transfect into PHD2-KO to monitor HIF1a transcriptional activity via a luciferase assay. All mutants show impaired activation. Some mutants also impaired stability in pulse chase turnover assays (except A228S, P317R, and F366L). In vitro purified PHD2 mutants display a minor loss in thermal stability and some propensity to aggregate. Using MST technology, they show that P317R is strongly impaired in binding to HIF1a and HIF2a, whereas other mutants are only slightly affected. Using NMR, they show that the PHD2 P317R mutation greatly reduces hydroxylation of P402 (HIF1a NODD), as well as P562 (HIF1a CODD), but to a lesser extent. Finally, BLI shows that the P317R mutation reduces affinity for CODD by 3-fold, but not NODD.

Strengths:

(1) Simple, easy-to-follow manuscript. Generally well-written.

(2) Disease-relevant mutations are studied in PHD2 that provide insights into its mechanism of action.

(3) Good, well-researched background section.

Weaknesses:

(1) Poor use of existing structural data on the complexes of PHD2 with HIF1a peptides and various metals and substrates. A quick survey of the impact of these mutations (as well as analysis by Chowdhury et al, 2016) on the structure and interactions between PHD2 peptides of HIF1a shows that the P317R mutation interferes with peptide binding. By contrast, F366L will affect the hydrophobic core, and A228S is on the surface, and it's not obvious how it would interfere with the stability of the protein.

(2) To determine aggregation and monodispersity of the PHD2 mutants using size-exclusion chromatography (SEC), equal quantities of the protein must be loaded on the column. This is not what was done. As an aside, the colors used for the SEC are very similar and nearly indistinguishable.

(3) The interpretation of some mutants remains incomplete. For A228S, what is the explanation for its reduced activity? It is not substantially less stable than WT and does not seem to affect peptide hydroxylation.

(4) The interpretation of the NMR prolyl hydroxylation is tainted by the high concentrations used here. First of all, there is a likely a typo in the method section; the final concentration of ODD is likely 0.18 mM, and not 0.18 uM (PNAS paper by the same group in 2024 reports using a final concentration of 230 uM). Here, I will assume the concentration is 180 uM. Flashman et al (JBC 2008) showed that the affinity of the NODD site (P402; around 10 uM) for PHD2 is 10-fold weaker than CODD (P564, around 1 uM). This likely explains the much faster kinetics of hydroxylation towards the latter. Now, using the MST data, let's say the P317R mutation reduces the affinity by 40-fold; the affinity becomes 400 uM for NODD (above the protein concentration) and 40 uM for CODD (below the protein concentration). Thus, CODD would still be hydroxylated by the P317R mutant, but not NODD.

(5) The discrepancy between the MST and BLI results does not make sense, especially regarding the P317R mutant. Based on the crystal structures of PHD2 in complex with the ODD peptides, the P317R mutation should have a major impact on the affinity, which is what is reported by MST. This suggests that the MST is more likely to be valid than BLI, and the latter is subject to some kind of artefact. Furthermore, the BLI results are inconsistent with previous results showing that PHD2 has a 10-fold lower affinity for NODD compared to CODD.

(6) Overall, the study provides some insights into mutants inducing erythrocytosis, but the impact is limited. Most insights are provided on the P317R mutant, but this mutant had already been characterized by Chowdhury et al (2016). Some mutants affect the stability of the protein in cells, but then no mechanism is provided for A228S or F366L, which have stabilities similar to WT, yet have impaired HIF1a activation.

Comments on revision:

While the authors have addressed my concerns regarding the SEC experiments and the structural interpretation of most mutants, I remain unconvinced by their interpretation of the P317R mutant and affinity measurements. The BLI and MST data remain inconsistent for P317R binding to CODD, and the authors' response is essentially that the fluorescent labeling of P317R (but not other mutants) uniquely interferes with binding to the NODD/CODD peptides, which does not make a lot of sense. The fluorescent labeling target lysine residues; while there are lysine in PHD2 in proximity to the peptide binding site, labeling these sites would affect binding to all mutants, not only P317R (which does not introduce any new labeling site). Furthermore, the authors did not really address the discrepancy with the observations by Flashman et al (2008) that NODD binds more weakly than CODD, which is inconsistent with their BLI results. Another point that makes me doubt the validity of the BLI results is the poor fit of the sensorgrams and the slow dissociation kinetics, which is inconsistent with the relatively low affinity in the 2-6 uM range.

---

## [Referee Report · Reviewer #2 (Public review)]

Summary:

Mutations in the prolyl hydroxylase, PHD2, cause erythrocytosis and, in some cases, can result in tumorigenesis. Taber and colleagues test the structural and functional consequences of seven patient-derived missense mutations in PHD2 using cell-based reporter and stability assays, and multiple biophysical assays, and find that most mutations are destabilizing. Interestingly, they discover a PHD2 mutant that can hydroxylate the C-terminal ODD, but not the N-terminal ODD, which suggests the importance of N-terminal ODD for biology. A major strength of the manuscript is the multidisciplinary approach used by the authors to characterize the functional and structural consequences of the mutations. However, the manuscript had several major weaknesses, such as an incomplete description of how the NMR was performed, a justification for using neighboring residues as a surrogate for looking at prolyl hydroxylation directly, or a reference to the clinical case studies describing the phenotypes of patient mutations. Additionally, the experimental descriptions for several experiments are missing descriptions of controls or validation, which limits their strength in supporting the claims of the authors.

Strengths:

(1) This manuscript is well-written and clear.

(2) The authors use multiple assays to look at the effects of several disease-associated mutations, which support the claims.

(3) The identification of P317R as a mutant that loses activity specifically against NODD, which could be a useful tool for further studies in cells.

Weaknesses:

Major:

(1) The source data for the patient mutations (Figure 1) in PHD2 is not referenced, and it's not clear where this data came from or if it's publicly available. There is no section describing this in the methods.

(2) The NMR hydroxylation assay.

A. The description of these experiments is really confusing. The authors have published a recent paper describing a method using 13C-NMR to directly detect proly-hydroxylation over time, and they refer to this manuscript multiple times as the method used for the studies under review. However, it appears the current study is using 15N-HSQC-based experiments to track the CSP of neighboring residues to the target prolines, so not the target prolines themselves. The authors should make this clear in the text, especially on page 9, 5th line, where they describe proline cross-peaks and refer to the 15N-HSQC data in Figure 5B.

B. The authors are using neighboring residues as reporters for proline hydroxylation, without validating this approach. How well do CSPs of A403 and I566 track with proline hydroxylation? Have the authors confirmed this using their 13C-NMR data or mass spec?

C. Peak intensities. In some cases, the peak intensities of the end point residue look weaker than the peak intensities of the starting residue (5B, PHD2 WT I566, 6 ct lines vs. 4 ct lines). Is this because of sample dilution (i.e., should happen globally)? Can the authors comment on this?

(3) Data validating the CRISPR KO HEK293A cells is missing.

(4) The interpretation of the SEC data for the PHD2 mutants is a little problematic. Subtle alterations in the elution profiles may hint at different hydrodynamic radii, but as the samples were not loaded at equal concentrations or volumes, these data seem more anecdotal, rather than definitive. Repeating this multiple times, using matched samples, followed by comparison with standards loaded under identical buffer conditions, would significantly strengthen the conclusions one could make from the data.

Minor:

(1) Justification for picking the seven residues is not clearly articulated. The authors say they picked 7 mutants with "distinct residue changes", but no further rationale is provided.

(2) A major finding of the paper is that a disease-associated mutation, P317R, can differentially affect HIF1 prolyhydroxylation, however, additional follow-up studies have not been performed to test this in cells or to validate the mutant in another method. Is it the position of the proline within the catalytic core, or the identity of the mutation that accounts for the selectivity?

Comments on revision:

The revised manuscript addresses most of my concerns, i.e performing SEC experiments under matched sample concentrations, and incorporating additional data to justify the use of surrogate residues to monitor proline hydroxylation. I appreciate the improvements in the text to clarify the NMR experiments, but I still find their description confusing. Although the authors are using neighboring residues to monitor proline hydroxylation (which they justify convincingly using supplementary data), the language in the text suggests they are (and can?) monitor them directly (i.e. referring to proline cross-peaks in an 15N-HSQC spectrum). The axis labels in Figure 5B also seem to have become mislabeled in this revised version.

---

## [Referee Report · Reviewer #3 (Public review)]

Summary:

This is an interesting and clinically relevant in vitro study by Taber et al., exploring how mutations in PHD2 contribute to erythrocytosis and/or neuroendocrine tumors. PHD2 regulates HIFα degradation through prolyl-hydroxylation, a key step in the cellular oxygen-sensing pathway.

Using a time-resolved NMR-based assay, the authors systematically analyze seven patient-derived PHD2 mutants and demonstrate that all exhibit structural and/or catalytic defects. Strikingly, the P317R variant retains normal activity toward the C-terminal proline but fails to hydroxylate the N-terminal site. This provides the first direct evidence that N-terminal prolyl-hydroxylation is not dispensable, as previously thought.

The findings offer valuable mechanistic insight into PHD2-driven effects and refine our understanding of HIF regulation in hypoxia-related diseases.

Strengths:

The manuscript has several notable strengths. By applying a novel time-resolved NMR approach, the authors directly assess hydroxylation at both HIF1α ODD sites, offering a clear functional readout. This method allows them to identify the P317R variant as uniquely defective in NODD hydroxylation, despite retaining normal activity toward CODD, thereby challenging the long-held view that the N-terminal proline is biologically dispensable. The work significantly advances our understanding of PHD2 function and its role in oxygen sensing, and might help in the future interpretation and clinical management of associated erythrocytosis.

Weaknesses:

There is a lack of in vivo/ex vivo validation. This is actually required to confirm whether the observed defects in hydroxylation-especially the selective NODD impairment in P317R-are sufficient to drive disease phenotypes such as erythrocytosis.

The reliance on HRE-luciferase reporter assays may not reliably reflect the PHD2 function and highlights a limitation in the assessment of downstream hypoxic signaling.

The study clearly documents the selective defect of the P317R mutant, but the structural basis for this selectivity is not addressed through high-resolution structural analysis (e.g., cryo-EM).

Given the proposed central role of HIF2α in erythrocytosis, direct assessment of HIF2α hydroxylation by the mutants would have strengthened the conclusions.

Comments on revision:

The revised manuscript by Taber et al. addresses the key points raised during the review process in a comprehensive and appropriate manner. While some limitations remain, such as the lack of in vivo validation or direct HIF2α assessment, I agree with the authors that these are beyond the scope of the current in vitro-focused study. The authors' primary goal was to define the structural and functional defects caused by disease-associated PHD2 mutations. In this respect, the evidence they present is largely convincing and methodologically appropriate. Additional clarifications and an expanded discussion of the luciferase assay's limitations and the P317R structural context strengthen the manuscript further.

---

## [Author Response]

The following is the authors’ response to the original reviews.

**Reviewer #1 (Public review):**
Summary:Taber et al report the biochemical characterization of 7 mutations in PHD2 that induce erythrocytosis. Their goal is to provide a mechanism for how these mutations cause the disease. PHD2 hydroxylates HIF1a in the presence of oxygen at two distinct proline residues (P564 and P402) in the "oxygen degradation domain" (ODD). This leads to the ubiquitylation of HIF1a by the VHL E3 ligase and its subsequent degradation. Multiple mutations have been reported in the EGLN1 gene (coding for PHD2), which are associated with pseudohypoxic diseases that include erythrocytosis. Furthermore, 3 mutations in PHD2 also cause pheochromocytoma and paraganglioma (PPGL), a neuroendocrine tumour. These mutations likely cause elevated levels of HIF1a, but their mechanisms are unclear. Here, the authors analyze mutations from 152 case reports and map them on the crystal structure. They then focus on 7 mutations, which they clone in a plasmid and transfect into PHD2-KO to monitor HIF1a transcriptional activity via a luciferase assay. All mutants show impaired activation. Some mutants also impaired stability in pulse chase turnover assays (except A228S, P317R, and F366L). In vitro purified PHD2 mutants display a minor loss in thermal stability and some propensity to aggregate. Using MST technology, they show that P317R is strongly impaired in binding to HIF1a and HIF2a, whereas other mutants are only slightly affected. Using NMR, they show that the PHD2 P317R mutation greatly reduces hydroxylation of P402 (HIF1a NODD), as well as P562 (HIF1a CODD), but to a lesser extent. Finally, BLI shows that the P317R mutation reduces affinity for CODD by 3-fold, but not NODD.Strengths:(1) Simple, easy-to-follow manuscript. Generally well-written.(2) Disease-relevant mutations are studied in PHD2 that provide insights into its mechanism of action.(3) Good, well-researched background section.Weaknesses:(1) Poor use of existing structural data on the complexes of PHD2 with HIF1a peptides and various metals and substrates. A quick survey of the impact of these mutations (as well as analysis by Chowdhury et al, 2016) on the structure and interactions between PHD2 peptides of HIF1a shows that the P317R mutation interferes with peptide binding. By contrast, F366L will affect the hydrophobic core, and A228S is on the surface, and it's not obvious how it would interfere with the stability of the protein.

Thank you for the comment. We have further analyzed the mutations on the available PHD2 crystal structures in complex with HIFα to discern how these substitution mutations may impact PHD2 structure and function. This analysis has been added into the discussion.

(2) To determine aggregation and monodispersity of the PHD2 mutants using size-exclusion chromatography (SEC), equal quantities of the protein must be loaded on the column. This is not what was done. As an aside, the colors used for the SEC are very similar and nearly indistinguishable.

Agreed. We have performed an additional experiment as suggested by the reviewer to further assess aggregation and hydrodynamic size. The colors used in the graph were changed for clearer differentiation between samples.

(3) The interpretation of some mutants remains incomplete. For A228S, what is the explanation for its reduced activity? It is not substantially less stable than WT and does not seem to affect peptide hydroxylation.

We agree with the reviewer that the causal mechanism for some of the tested disease-causing mutants remain unclear. The negative findings also raise the notion, perhaps considered controversial, that there may be other substrates of PHD2 that are impacted by certain mutations, which contribute to disease pathogenesis. A brief paragraph discussing this has been included in the discussion.

(4) The interpretation of the NMR prolyl hydroxylation is tainted by the high concentrations used here. First of all, there is a likely a typo in the method section; the final concentration of ODD is likely 0.18 mM, and not 0.18 uM (PNAS paper by the same group in 2024 reports using a final concentration of 230 uM). Here, I will assume the concentration is 180 uM. Flashman et al (JBC 2008) showed that the affinity of the NODD site (P402; around 10 uM) for PHD2 is 10-fold weaker than CODD (P564, around 1 uM). This likely explains the much faster kinetics of hydroxylation towards the latter. Now, using the MST data, let's say the P317R mutation reduces the affinity by 40-fold; the affinity becomes 400 uM for NODD (above the protein concentration) and 40 uM for CODD (below the protein concentration). Thus, CODD would still be hydroxylated by the P317R mutant, but not NODD.

The HIF1α concentration was indeed an oversight, which will be corrected to 0.18 mM. The study by Flashman et al.[1] showing PHD2 having a lower affinity to the NODD than CODD likely contributes to the differential hydroxylation rates via PHD2 WT. We showed here via MST that PHD2 P317R had K[d] of 320 ± 20 uM for HIF1αCODD, which should have led to a severe enzymatic defect, even at the high concentrations used for NMR (180 uM). However, we observed only a subtle reduction in hydroxylation efficiency in comparison to PHD2 WT. Thus, we performed another binding method using BLI that showed a mild binding defect on CODD by PHD2 P317R, consistent with NMR data. The perplexing result is the WT-like binding to the NODD by PHD2 P317R, which appears inconsistent with the severe defect in NODD hydroxylation via PHD2 P317R as measured via NMR. These results suggest that there are supporting residues within the PHD2/NODD interface that help maintain binding to NODD but compromise the efficiency of NODD hydroxylation upon PHD2 P317R mutation.

(5) The discrepancy between the MST and BLI results does not make sense, especially regarding the P317R mutant. Based on the crystal structures of PHD2 in complex with the ODD peptides, the P317R mutation should have a major impact on the affinity, which is what is reported by MST. This suggests that the MST is more likely to be valid than BLI, and the latter is subject to some kind of artefact. Furthermore, the BLI results are inconsistent with previous results showing that PHD2 has a 10-fold lower affinity for NODD compared to CODD.

The reviewer’s structural prediction that P317R mutation should cause a major binding defect, while agreeable with our MST data, is incongruent with our NMR and the data from Chowdhury et al.[2] that showed efficient hydroxylation of CODD via PHD2 P317R. Moreover, we have attempted to model NODD and CODD on apo PHD2 P317R structure and found that the mutation had no major impact on CODD while the mutated residue could clash with NODD, causing a shifting of peptide positioning on the protein. However, these modeling predictions, like any in silico projections, would need experimental validation. As mentioned in our preceding response, we also performed BLI, which showed that PHD2 P317R had a minor binding defect for CODD, consistent with the NMR results and findings by Chowdhury et al[2]. NODD binding was also measured with BLI as purified NODD peptides were not amenable for soluble-based MST assay, which showed similar K[d]’s for PHD2 WT and P317R. Considering the absence of NODD hydroxylation via PHD2 P317R as measured by NMR and modeling on apo PHD2 P317R, we posit that P317R causes deviation of NODD from its original orientation that may not affect binding due to the other interactions from the surrounding elements but unfortunately disallows NODD from turnover. Further study would be required to validate such notion, which we feel is beyond the scope of this manuscript.

(6) Overall, the study provides some insights into mutants inducing erythrocytosis, but the impact is limited. Most insights are provided on the P317R mutant, but this mutant had already been characterized by Chowdhury et al (2016). Some mutants affect the stability of the protein in cells, but then no mechanism is provided for A228S or F366L, which have stabilities similar to WT, yet have impaired HIF1a activation.

We thank the reviewer for raising these and other limitations. We have expanded on the shortcomings of the present study but would like to underscore that the current work using the recently described NMR assay along with other biophysical analyses suggests a previously under-appreciated role of NODD hydroxylation in the normal oxygen-sensing pathway.

**Reviewer #2 (Public review):**
Summary:Mutations in the prolyl hydroxylase, PHD2, cause erythrocytosis and, in some cases, can result in tumorigenesis. Taber and colleagues test the structural and functional consequences of seven patientderived missense mutations in PHD2 using cell-based reporter and stability assays, and multiple biophysical assays, and find that most mutations are destabilizing. Interestingly, they discover a PHD2 mutant that can hydroxylate the C-terminal ODD, but not the N-terminal ODD, which suggests the importance of N-terminal ODD for biology. A major strength of the manuscript is the multidisciplinary approach used by the authors to characterize the functional and structural consequences of the mutations. However, the manuscript had several major weaknesses, such as an incomplete description of how the NMR was performed, a justification for using neighboring residues as a surrogate for looking at prolyl hydroxylation directly, or a reference to the clinical case studies describing the phenotypes of patient mutations. Additionally, the experimental descriptions for several experiments are missing descriptions of controls or validation, which limits their strength in supporting the claims of the authors.Strengths:(1) This manuscript is well-written and clear.(2) The authors use multiple assays to look at the effects of several disease-associated mutations, which support the claims.(3) The identification of P317R as a mutant that loses activity specifically against NODD, which could be a useful tool for further studies in cells.Weaknesses:Major:(1) The source data for the patient mutations (Figure 1) in PHD2 is not referenced, and it's not clear where this data came from or if it's publicly available. There is no section describing this in the methods.

Clinical and patient information on disease-causing PHD2 mutants was compiled from various case reports and summarized in an excel sheet found in the Supplementary Information. The case reports are cited in this excel file. A reference to the supplementary data has been added to the Figure 1 legend and in the introduction.

(2) The NMR hydroxylation assay.A. The description of these experiments is really confusing. The authors have published a recent paper describing a method using 13C-NMR to directly detect proly-hydroxylation over time, and they refer to this manuscript multiple times as the method used for the studies under review. However, it appears the current study is using 15N-HSQC-based experiments to track the CSP of neighboring residues to the target prolines, so not the target prolines themselves. The authors should make this clear in the text, especially on page 9, 5th line, where they describe proline cross-peaks and refer to the 15N-HSQC data in Figure 5B.

As the reviewer mentioned, the assay that we developed directly measures the target proline residues. This assay is ideal when mutations near the prolines are studied, such as A403, Y565 (He et al[3]). In this previous work, we observed that the shifting of the target proline cross-peaks due to change in electronegativity on the pyrrolidine ring of proline in turn impacted the neighboring residues[3], which meant that the neighboring residues can be used as reporter residues for certain purposes. In this study, we focused on investigating the mutations on PHD2 while leaving the sequence of the HIF-1α unchanged by using solely 15N-HSQC-based experiments without the need for double-labeled samples. Nonetheless, we thank the reviewer for pointing out the confusion in the text and we have corrected and clarified our description of this assay.

B. The authors are using neighboring residues as reporters for proline hydroxylation, without validating this approach. How well do CSPs of A403 and I566 track with proline hydroxylation? Have the authors confirmed this using their 13C-NMR data or mass spec?

For previous studies, we performed intercalated 15N-HSQC and 13C-CON experiments for the kinetic measurements of wild-type HIF-1α and mutants. We observed that the shifting pattern of A403 and I566 in the 15N-HSQC spectra aligned well with the ones of P402 and P564, respectively, in the 13C-CON spectra. Representative data has been added to Supplemental Data.

C. Peak intensities. In some cases, the peak intensities of the end point residue look weaker than the peak intensities of the starting residue (5B, PHD2 WT I566, 6 ct lines vs. 4 ct lines). Is this because of sample dilution (i.e., should happen globally)? Can the authors comment on this?

This is an astute observation by the reviewer. We checked and confirmed that for all kinetic datasets, the peak intensities of the end point residue are always slightly lower than the ones of the starting. This includes the cases for PHD2 A228S and P317R in 5B, although not as obvious as the one of PHD2 WT. We agree with the reviewer that the sample dilution is a factor as a total volume of 16 microliters of reaction components was added to the solution to trigger the reaction after the first spectrum was acquired. It is also likely that rate of prolyl hydroxylation becomes extremely slow with only a low amount of substrate available in the system. Therefore, the reaction would not be 100% complete which was detected by the sensitive NMR experimentation.

(3) Data validating the CRISPR KO HEK293A cells is missing.

We thank the reviewer for noting this oversight. Western blots validating PHD2 KO in HEK293A cells have been added to the Supplementary Data file.

(4) The interpretation of the SEC data for the PHD2 mutants is a little problematic. Subtle alterations in the elution profiles may hint at different hydrodynamic radii, but as the samples were not loaded at equal concentrations or volumes, these data seem more anecdotal, rather than definitive. Repeating this multiple times, using matched samples, followed by comparison with standards loaded under identical buffer conditions, would significantly strengthen the conclusions one could make from the data.

Agreed. We have performed an additional experiment as suggested with equal volume and concentration of each PHD2 construct loaded onto the SEC column for better assessment of aggregation. Notably, our conclusion remained unchanged.

Minor:(1) Justification for picking the seven residues is not clearly articulated. The authors say they picked 7 mutants with "distinct residue changes", but no further rationale is provided.

Additional justification for the selection of the mutants has been added to the ‘Mutations across the PHD2 enzyme induce erythrocytosis’ section. Briefly, some mutants were chosen based on their frequency in the clinical data and their presence in potential mutational hot spots. Various mutations were noted at W334 and R371, while F366L was identified in multiple individuals. Additionally, 9 cases of PHD2-driven disease were reported to be caused from mutations located between residues 200 to 210 while 13 cases were reported between residues 369-379, so G206C and R371H were chosen to represent potential hot spots. To examine a potential genotype-phenotype relationship, two of the mutants responsible for neuroendocrine tumor development, A228S and H374R, were also selected. Finally, mutations located close or on catalytic core residues (P317R, R371H, and H374R) were chosen to test for suspected defects.

(2) A major finding of the paper is that a disease-associated mutation, P317R, can differentially affect HIF1 prolyhydroxylation, however, additional follow-up studies have not been performed to test this in cells or to validate the mutant in another method. Is it the position of the proline within the catalytic core, or the identity of the mutation that accounts for the selectivity?

This is the very question that we are currently addressing but as a part of a follow-up study. Indeed, one thought is that the preferential defect observed could be the result of the loss of proline, an exceptionally rigid amino acid that makes contact with the backbone twice, or the addition of a specific amino acid, namely arginine, a flexible amino acid with an added charge at this site. Although beyond the scope of this manuscript, we will investigate whether such and other characteristics in this region of PHD2/HIF1α interface contribute to the differential hydroxylation.

**Reviewer #3 (Public review):**
Summary:This is an interesting and clinically relevant in vitro study by Taber et al., exploring how mutations in PHD2 contribute to erythrocytosis and/or neuroendocrine tumors. PHD2 regulates HIFα degradation through prolyl-hydroxylation, a key step in the cellular oxygen-sensing pathway.Using a time-resolved NMR-based assay, the authors systematically analyze seven patient-derived PHD2 mutants and demonstrate that all exhibit structural and/or catalytic defects. Strikingly, the P317R variant retains normal activity toward the C-terminal proline but fails to hydroxylate the N-terminal site. This provides the first direct evidence that N-terminal prolyl-hydroxylation is not dispensable, as previously thought.The findings offer valuable mechanistic insight into PHD2-driven effects and refine our understanding of HIF regulation in hypoxia-related diseases.Strengths:The manuscript has several notable strengths. By applying a novel time-resolved NMR approach, the authors directly assess hydroxylation at both HIF1α ODD sites, offering a clear functional readout. This method allows them to identify the P317R variant as uniquely defective in NODD hydroxylation, despite retaining normal activity toward CODD, thereby challenging the long-held view that the N-terminal proline is biologically dispensable. The work significantly advances our understanding of PHD2 function and its role in oxygen sensing, and might help in the future interpretation and clinical management of associated erythrocytosis.Weaknesses:(1) There is a lack of in vivo/ex vivo validation. This is actually required to confirm whether the observed defects in hydroxylation-especially the selective NODD impairment in P317R-are sufficient to drive disease phenotypes such as erythrocytosis.

We thank the reviewer for this comment, and while we agree with this statement, the objective of this study per se was to elucidate the structural and/or functional defect caused by the various diseaseassociated mutations on PHD2. The subsequent study would be to validate whether the identified defects, in particular the selective NODD impairment, would lead to erythrocytosis in vivo. However, we feel that such study would be beyond the scope of this manuscript.

(2) The reliance on HRE-luciferase reporter assays may not reliably reflect the PHD2 function and highlights a limitation in the assessment of downstream hypoxic signaling.

Agreed. All experimental assays and systems have limitations. The HRE-luciferase assay used in the present manuscript also has limitations such as the continuous expression of exogenous PHD2 mutants driven via CMV promoter. Thus, we performed several additional biophysical methodologies to interrogate the disease-causing PHD2 mutants. The limitations of the luciferase assay have been expanded in the revised manuscript.

(3) The study clearly documents the selective defect of the P317R mutant, but the structural basis for this selectivity is not addressed through high-resolution structural analysis (e.g., cryo-EM).

We thank the reviewer for the comment. While solving the structure of PHD2 P317R in complex with HIFα substrate is beyond the scope for this study, a structure of PHD2 P317R in complex with a clinically used inhibitor has been solved (PDB:5LAT). In analyzing this structure and that of PHD2 WT in complex with NODD, Chowdhury et al[2] stated that P317 makes hydrophobic contacts with LXXLAP motif on HIFα and R317 is predicted to interact differently with this motif. While this analysis does not directly elucidate the reason for the preferential NODD defect, it supports the possibility that P317R substitution may be more detrimental for enzymatic activity on NODD than CODD. We have discussed this notion in the revised manuscript.

(4) Given the proposed central role of HIF2α in erythrocytosis, direct assessment of HIF2α hydroxylation by the mutants would have strengthened the conclusions.

We thank the reviewer for this comment, but we feel that such study would be beyond the scope of the present study. We observed that the PHD2 binding patterns to HIF1α and HIF2α were similar, and we have previously assigned >95% of the amino acids in HIF1α ODD for NMR study[3]. Thus, we first focused on the elucidation of possible defects on disease-associated PHD2 mutants using HIF1α as the substrate with the supposition that an identified deregulation on HIF1α could be extended to HIF2α paralog. However, we agree with the reviewer that future studies should examine the impact of PHD2 mutants directly on HIF2α.

References:

(1) Flashman, E. et al. Kinetic rationale for selectivity toward N- and C-terminal oxygen-dependent degradation domain substrates mediated by a loop region of hypoxia-inducible factor prolyl hydroxylases. J Biol Chem 283, 3808-3815 (2008).

(2) Chowdhury, R. et al. Structural basis for oxygen degradation domain selectivity of the HIF prolyl hydroxylases. Nat Commun 7, 12673 (2016).

(3) He, W., Gasmi-Seabrook, G.M.C., Ikura, M., Lee, J.E. & Ohh, M. Time-resolved NMR detection of prolyl-hydroxylation in intrinsically disordered region of HIF-1alpha. Proc Natl Acad Sci U S A 121, e2408104121 (2024).

**Reviewer #1 (Recommendations for the authors):**
(1) To increase the impact and significance of this work, I would recommend determining the mechanism by which A228S and F366L impair PHD2. Are these mutations affecting interactions with proteins other than HIF1a? Furthermore, does the F366L mutation affect the hydroxylation rate? This should be measured. The authors should also perform a more in-depth structural analysis of these mutations and perhaps use AlphaFold to identify how these sites may be involved in other interactions.

We thank the reviewer for the recommendations. A paragraph discussing the quandary of A228S and F366L has been added to the discussion as well as an in-depth structural analysis of each selected mutant. While AlphaFold is excellent at predicting protein structures overall, its capability to predict the effect of single point mutation, such as those in this study, is limited. Therefore, it was not utilized for this paper.

(2) For the aggregation assay, I recommended injecting the same quantity of protein on the SEC. If the aggregation-prone mutants' yields were too low, then reduced amounts of the other mutants should be injected.

Agreed. An additional experiment was performed in which similar concentrations of each mutant protein was loaded onto the SEC column and chromatograms was normalized according to the molecular concentration. Results from this experiment have been added to replace the previously performed aggregation assay. Notably, the data from the revised experiment did not change the outcome or conclusion of the study.

(3) For the NMR kinetics data, the authors should discuss the impact of affinities and concentrations on the reaction rate and incorporate this analysis framework to interpret their data.

Done. As discussed in depth in response to Public Reviewer 1’s fourth comment, we observed only a subtle reduction in hydroxylation efficiency of HIF1aCODD by PHD2 P317R in comparison to PHD2 WT. Upon performing BLI, we found PHD2 P317R displays only a mild binding defect on the CODD and NODD. The WT-like binding to the NODD by PHD2 P317R appears to be inconsistent with the severe defect in NODD hydroxylation via PHD2 P317R as measured via NMR. These results suggest that there are supporting residues within the PHD2/NODD interface that help maintain binding to NODD but compromise the efficiency of NODD hydroxylation upon PHD2 P317R mutation.

**Reviewer #2 (Recommendations for the authors):**
It is unclear where the source data came from describing the patient mutations, or if it is publicly available. Several minor issues were noted with several of the figures or methods:(1) Figure 2C. It is not clear what data are being compared for significance. The lines don't seem to clearly distinguish this.

Done. The significance lines have been adjusted in the figure to better convey which data are being compared.

(2) Please incorporate the calculated biophysical constants (KD, TM, etc, average +/- std dev) from the tables into the figures or figure legends that show the data from which they are calculated.

Done. References to the corresponding tables have been added to the appropriate figure legends.

(3) Figure 3C, the data for F366L do not appear normalized in the same way as the other constructs.

CD melt values for F366L were normalized in the same way as other constructs but due to noisier data acquired between 25-37°C, the top value of the sigmoidal curve is slightly higher than the other constructs (F366L: 1.066, WT: 1.007, A228S: 1.000, P317R: 1.015, R371H: 1.005).

(4) For Figure 1B, it would be helpful to highlight the mutants characterized in the current study with a different color/symbol to help show the number of cases.

Done. Dots representing the selected mutants have been highlighted in red in Figure 1B.

(5) A description of the isotopic labeling of PHD2 is missing from the methods.

Due to the nature of the NMR assay, no isotopic labeling was required for PHD2.

**Reviewer #3 (Recommendations for the authors):**
(1) To further strengthen the manuscript, the authors could consider exploring the relevance of their in vitro findings in a more physiological context.

We thank the reviewer for the suggestion, and we will certainly consider furthering our investigation in a more physiological context for future studies.

(2) If technically feasible, integrating direct analyses of HIF2α regulation by the PHD2 mutants would better reflect the clinical phenotype, given the known importance of HIF2α in erythrocytosis.

We agree that HIF2α is important in the context of erythrocytosis, but through MST we observed no difference in binding pattern between HIF1 and HIF2 and the selected PHD2 mutants. As we had previously assigned >95% of residues for HIF1α ODD for NMR assay, we analyzed HIF1 with the supposition that any defects observed would likely apply to HIF2. However, we agree that future studies on the impact of PHD2 mutants directly on HIF2 would be beneficial to supplement our understanding of pseudohypoxic disease.

(3) Additionally, although perhaps more suitable for future work or discussion, structural modeling or highresolution structural studies of the P317R variant could offer valuable insight into the observed NODD selectivity defect.

We thank the reviewer for the suggestion. While solving the structure of PHD2 P317R in complex with NODD is beyond the scope of this manuscript, a crystal structure of PHD2 P317R in complex with an inhibitor has been solved and insights from this structure have been added to the discussion.

(4) Finally, a brief clarification or discussion of the limitations of the luciferase reporter assay-especially in the context of aggregation-prone mutants-would help readers better interpret the functional data.

We thank the reviewer for the suggestion. The limitations of the luciferase reporter assay in regard to its inability to detect defects with aggregation-prone mutants have been elaborated on in the discussion.